# The endocannabinoid anandamide is an airway relaxant in health and disease

Annika Simon[1], Thomas von Einem[2], Alexander Seidinger[1], Michaela Matthey[1], Laura Bindila [3] & Daniela Wenzel [1,2] ✉

Chronic obstructive airway diseases are a global medical burden that is expected to increase in the near future. However, the underlying mechanistic processes are poorly understood so far. Herein, we show that the endocannabinoid anandamide (AEA) induces prominent airway relaxation in vitro and in vivo. In contrast to 2-arachidonlyglycerol-induced airway relaxation, this is mediated by fatty acid amide hydrolase (FAAH)-dependent metabolites. In particular, we identify mouse and also human epithelial and airway smooth muscle cells as source of AEA-induced prostaglandin E2 production and cAMP as direct mediator of AEA-dependent airway relaxation. Mass spectrometry experiments demonstrate reduced levels of endocannabinoid-like compounds in lungs of ovalbumin-sensitized mice indicating a pathophysiological relevance of endocannabinoid signalling in obstructive airway disease. Importantly, AEA inhalation protects against airway hyper-reactivity after ovalbumin sensitization. Thus, this work highlights the AEA/FAAH axis as a critical regulator of airway tone that could provide therapeutic targets for airway relaxation.

Endocannabinoids are endogenously produced lipophilic compounds that can exert similar effects as tetrahydrocannabinol (THC), the main constituent of the cannabis plant. Originally, they had been identified as important regulators in the central and peripheral nervous system affecting memory, appetite and pain sensation[1]. Later, it has become clear that endocannabinoids also modulate the homeostasis of other organs. High concentrations of endocannabinoids have been detected in the lung[2,3]. In this organ endocannabinoids also display functional effects on vessels as anandamide (AEA) and 2-arachidonlyglycerol (2-AG) were demonstrated to control pulmonary arterial tone. In isolated pulmonary arteries (PAs) of different species endocannabinoids were reported to induce vasorelaxation[4–7]. Interestingly, in murine and rabbit[2] whole lung preparations ex vivo as well as in mouse in vivo[8], we and others found a strong pulmonary vasoconstriction by AEA mediated by fatty acid amide hydrolase (FAAH). Our experiments revealed that this effect was specific for AEA and not found for 2-AG. In addition, we could show that this response is of physiological relevance as

FAAH-dependent AEA degradation proved to be a central mechanism of hypoxia-dependent vasoconstriction[8]. Because of its important physiological function in pulmonary vessels we then wondered if AEA can also affect airways and if this is of pathophysiological relevance. A previous study provided hints that endocannabinoids may influence cough and airway tone under very special circumstances[3] but it is still unclear how a potential tone regulatory effect is mediated and if this pathway provides therapeutic targets. Current treatment regimens for obstructive airway disease mainly rely on airway relaxation by beta2 adrenergic agonists. However, this strategy is limited since long term use of these compounds results in desensitization or even induces adverse effects[9]. Thus, there is an urgent need for alternative pharmacological strategies for obstructive airway disease. In this study we can show by various in vitro, ex vivo and in vivo approaches that AEA is a strong airway relaxant via FAAH-dependent arachidonic acid (AA), PGE2 and cAMP generation in mouse and human. The compound can be locally applied via

[1]Department of Systems Physiology, Medical Faculty, Ruhr University of Bochum, Bochum, Germany. [2]Institute of Physiology I, Life&Brain Center, Medical Faculty, University of Bonn, Bonn, Germany. [3]Institute of Physiological Chemistry, University Medical Center of the Johannes Gutenberg University of Mainz, Mainz, Germany. ✉e-mail: daniela.wenzel@rub.de

inhalation and does not induce cardiovascular side effects nor desensitization. Moreover, it prevents airway constriction not only in healthy but also in ovalbumin (OVA)-sensitized mice. Thus, this study highlights the AEA/FAAH pathway as a central regulator of airway tone.

## Results

### Anandamide induces airway relaxation

First, we examined the effect of anandamide (AEA) on airway tone in isometric force measurements of mouse tracheal rings in a wire-myograph. When applied to baseline force, single dose application of AEA (10 μM) had no effect (Supplementary Fig. 1a). In contrast, after pre-constriction single dose application of AEA (10 μM) induced a very strong airway relaxation (Fig. 1a, c) irrespective of the type of pre-constrictor (Supplementary Fig. 1b). This relaxation was not found in response to the solvent ethanol (EtOH) (Fig. 1b, c). Relaxation by AEA was dose-dependent starting at 10 nM (Supplementary Fig. 1c and Fig. 1d) which is in the range of AEA concentrations reported for human plasma[10].

### AEA-induced airway relaxation is mediated via FAAH-dependent metabolites

Then, we determined the molecular mechanism by which AEA evokes airway relaxation. In previous studies most effects of AEA have been attributed to its binding to CB1 and CB2 receptors. In contrast, we have recently shown that AEA degradation via the enzyme fatty acid amide hydrolase (FAAH) plays a key role in hypoxic pulmonary constriction of pulmonary vessels[8]. In mouse trachea we now found low CB1 but distinct CB2 and FAAH expression (Supplementary Fig. 1d). Our isometric force measurements revealed that AEA-induced airway

relaxation was independent from CB1 and CB2 receptors because it was found to be unaltered in Cnr1/2−/− mice (Fig. 1c). These findings suggest that FAAH-dependent metabolites are responsible for AEA-induced airway relaxation. In order to test this we pretreated WT mice with the pharmacological FAAH inhibitor URB597 or examined FAAH−/− mice in isometric force measurements. We found that this could almost completely prevent airway relaxation by AEA (Fig. 1c). The important role of FAAH-dependent metabolites for airway relaxation by AEA was corroborated by experiments with application of the non-hydrolyzable AEA analog methanandamide (Met-AEA) that did not affect airway tone either (Fig. 1c). To identify the cell type/s that are responsible for AEA-dependent airway relaxation we have mechanically removed the epithelium in tracheal rings (Supplementary Fig. 1e–h) and then performed isometric force measurements. This treatment resulted in an airway relaxation that was diminished by ~60% compared to preparations with intact epithelium (Fig. 1e) indicating that there is an epithelium-dependent as well as an epithelium-independent component of airway relaxation by AEA. This is also corroborated by FAAH immunostainings revealing FAAH expression in the epithelium but also in alpha smooth muscle actin (asmac)+ smooth muscle cells of trachea (Fig. 1f). Thus, AEA appears to evoke airway relaxation via FAAH-dependent metabolites that are generated in the epithelium and smooth muscle layer of airways. As N-acylethanolamine acid amide hydrolase (NAAA) is an alternative enzyme that can cleave AEA, we analyzed if this also plays a role in airway relaxation by AEA. Preincubation of tracheal rings with the NAAA inhibitor ARN726 (ARN, 10 μM) did not affect AEA-dependent bronchorelaxation (Supplementary Fig. 1i). Thus, we conclude that NAAA is not involved in the regulation of airway tone by AEA.

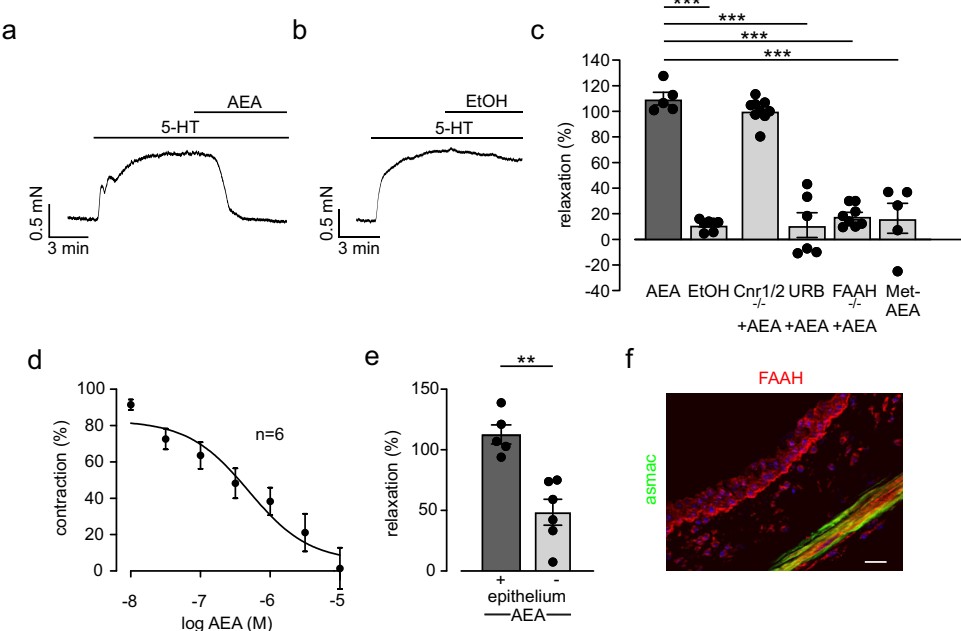

**Fig. 1 | Anandamide (AEA) induces airway relaxation via fatty acid amide hydrolase (FAAH) in mouse trachea ex vivo. a** Original trace of isometric force measurements in a myograph demonstrates strong tone decrease of tracheal ring from C57BL/6J mouse by a single dose of AEA (10 μM). **b** Original trace of isometric force measurements shows no effect by application of the solvent ethanol (EtOH). **c** Statistical analysis of airway tone in response to AEA indicates that AEA (n = 5) evokes relaxation independent from CB1 and CB2 receptors (Cnr1/Cnr2−/−, n = 9); URB597 (URB, 1 μM, n = 6), methanandamide (Met-AEA, 10 μM, n = 5); (EtOH, n = 7, FAAH−/−, n = 8). Measurements were performed in independent samples. One way ANOVA, Tukey's post hoc test (AEA vs EtOH ***p = 4.8 × 10⁻¹²; AEA vs URB ***p = 1.1 × 10⁻¹¹; AEA vs FAAH−/− ***p = 1.6 × 10⁻¹¹; AEA vs Met-AEA ***p = 1.5 × 10⁻¹⁰).

**d** Dose response curves of AEA from trachea of C57BL/6J mice (n = 6 independent samples) show that the AEA-dependent airway relaxation is dose-dependent. **e** Statistical analysis of isometric force measurements demonstrates that the extent of airway relaxation by AEA (10 μM) is different in tracheas with and without epithelium in C57BL/6J mice (+n = 5, −n = 6 independent samples), unpaired two-tailed Student's t test **p = 0.0012. **f** Immunostaining reveals FAAH expression in epithelial as well as smooth muscle cells of C57BL/6J mouse trachea (red = FAAH, green = alpha smooth muscle actin, staining was performed twice), scale bar = 20 μm. **c**–**e**) Data are presented as mean values ± SEM. Source data are provided as a Source Data file.

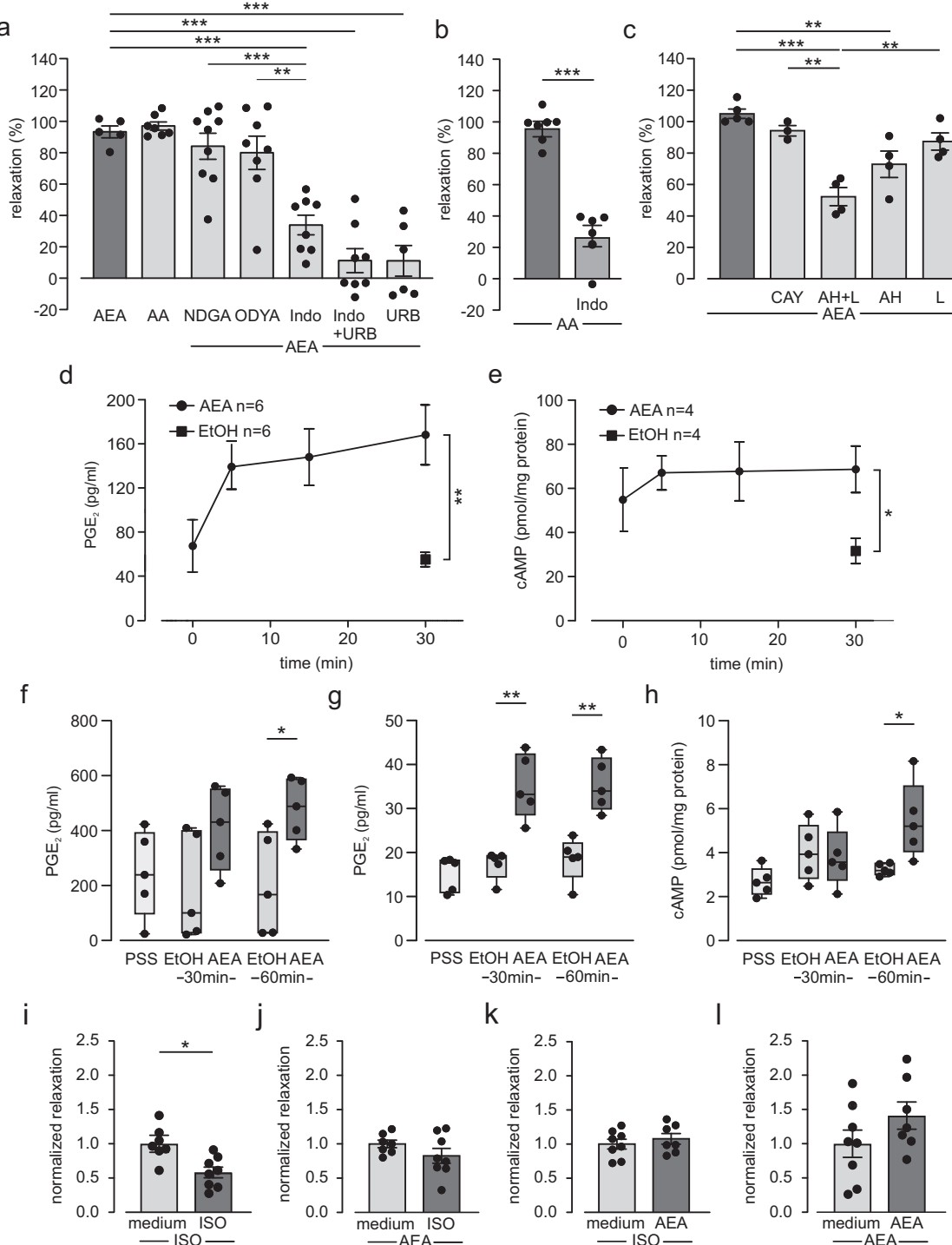

## AEA induces airway tone decrease by activation of EP2 and EP4 receptors

To investigate in detail the role of FAAH-dependent hydrolysis in AEA-induced airway relaxation we applied the direct AEA metabolite arachidonic acid (AA, 10 μM) in the wire-myograph. This compound induced a strong airway relaxation similar to AEA (Fig. 2a). We also tested ethanolamine, the FAAH-dependent metabolite that is produced together with AA when AEA is degraded. Application of ethanolamine (10 μM) alone on basal airway tone induced a small bronchoconstriction (0.6 ± 0.4 mN, n = 7). Because AA can be further metabolized by cyclooxygenases (COX), 5-lipoxygenase (5-LOX) and CYP450 enzymes we also applied the respective pharmacological inhibitors in isometric force measurements with AEA, all inhibitors

applied showed no effect on baseline tone. We found that the 5-LOX inhibitor nordihydroguaiaretic acid (NDGA, 10 μM) and the CYP450 inhibitor 17-octadecynoic acid (ODYA, 1 μM) had no effect, while the COX inhibitor indomethacin (Indo, 10 μM) strongly reduced AEA-dependent airway relaxation (Fig. 2a). These data suggest that AEA induces airway relaxation via its FAAH-dependent hydrolysis to AA and AA metabolites. COX can also directly metabolize AEA unrelated to FAAH-dependent hydrolysis to AA[11]. To corroborate that FAAH is involved in AEA-dependent relaxation we first combined inhibition of FAAH by URB597 and inhibition of COX by Indo and tested the bronchorelaxing effect by AEA. Our data reveal that the additional application of Indo did not further reduce AEA-dependent relaxation when compared with URB597 alone (Fig. 2a) indicating that COX is

**Fig. 2 | AEA induces airway relaxation by cyclooxydase (COX)-dependent prostaglandin E2 (PGE2) and cAMP generation. a** Statistical analysis of airway tone in tracheal rings from C57BL/6J mice indicates that AEA (10 μM, $n = 5$)-dependent airway relaxation is mediated via arachidonic acid (AA) and COX-dependent metabolites; arachidonic acid (AA, 10 μM, $n = 7$), nordihydroguaiaretic acid (NDGA, 10 μM; $n = 9$), 17-octadecynoic acid (ODYA, 1 μM, $n = 8$), indomethacin (Indo, 10 μM, $n = 8$), and Indo+URB597 ($n = 8$). Note that data for URB ($n = 6$) are taken from Fig. 1c. All results were derived from independent samples. One-way ANOVA, Tukey's post hoc test (AEA vs URB ***$p = 1.6 \times 10^{-6}$; AEA vs Indo+URB ***$p = 4.4 \times 10^{-7}$; AEA vs Indo ***$p = 2.3 \times 10^{-4}$; NDGA vs Indo ***$p = 2.5 \times 10^{-4}$; ODYA vs Indo **$p = 0.0014$). **b** Statistical analysis of airway tone in tracheal rings from C57BL/6J mice demonstrates strong inhibition of AA-dependent relaxation by Indo (AA, $n = 7$; Indo $n = 6$, independent samples). Unpaired two-tailed Student's $t$ test ***$p = 1.4 \times 10^{-6}$. **c** Statistical analysis of airway tone in tracheal rings from C57BL/6J mice reveals that AEA-induced airway relaxation is dependent on EP2 and EP4 receptors, AEA ($n = 5$), CAY10441 (CAY, 3 μM, $n = 3$), AH + L ($n = 4$), AH6809 (AH, 10 μM, $n = 4$), and L161,982 (L, 2 μM, $n = 4$). Results were derived from independent samples. One way ANOVA, Tukey's post hoc test (AEA vs AH **$p = 0.0047$; AEA vs AH + L ***$p = 3.2 \times 10^{-5}$; CAY vs AH + L **$p = 0.0014$; AH + L vs L **$p = 0.0035$). **d** AEA evokes PGE2 production in the supernatant of trachea (C57BL/6J mice, $n = 6$ independent samples) in a time-dependent manner. Unpaired two-tailed $t$ test with Welch correction **$p = 0.0079$. **e** AEA induces cAMP production in tracheal tissue from C57BL/6J mice ($n = 4$ independent samples) in a time-dependent manner. Unpaired two-tailed Student's $t$ test *$p = 0.022$. **f** AEA increases PGE2 production in the supernatant of human tracheal epithelial cells (hTEPC) after 60 min ($n = 5$ independent samples). Two way ANOVA, Bonferroni's post hoc test *$p = 0.037$. Values are expressed as box and whiskers plot, with boxes indicating the Q1 and Q3 ranges, center line representing the median and whiskers as minimum and maximum values. **g** AEA increases PGE2 production in human airway smooth muscle cells (hASMC) after 30 and 60 min ($n = 5$ independent samples). Two way ANOVA, Bonferroni's post hoc test (30 min: **$p = 3.5 \times 10^{-5}$, 60 min: **$p = 7.1 \times 10^{-5}$). Values are expressed as box and whiskers plot, with boxes indicating the Q1 and Q3 ranges, center line representing the median and whiskers as minimum and maximum values. **h** AEA increases cAMP production in hASMC after 60 min ($n = 5$ independent samples). Two way ANOVA, Bonferroni's post hoc test *$p = 0.012$. Values are expressed as box and whiskers plot, with boxes indicating the Q1 and Q3 ranges, center line representing the median and whiskers as minimum and maximum values. **i–l** Statistical analysis of airway relaxation by isoprenaline (ISO, 10 μM) (**i, k**) or AEA (10 μM) (**j, l**) after long-term treatment of tracheal rings from C57BL/6J mice with medium (left bar, $n = 7$ (**i, j**), $n = 8$ (**k, l**)), ISO (right bar, $n = 8$ (**i, j**)) or AEA (right bar, $n = 7$ (**k, l**)) reveals that prolonged ISO incubation reduces ISO-dependent but not AEA-dependent relaxation while AEA incubation has no effect. Unpaired two-tailed Student's $t$ test *$p = 0.012$ (**l**). **a–e, i–l** Data are presented as mean values ± SEM. Source data are provided as a Source Data file.

downstream of FAAH mediating this response. Next, we determined the effect of Indo on AA-induced bronchorelaxation to compare the effects of COX inhibition on AEA- and AA-induced bronchorelaxation. The results demonstrate that Indo reduces relaxation by AA to a similar extent as relaxation by AEA (Indo + AEA 33.9 ± 6.2% $n = 8$ (Fig. 2a); Indo +AA 27.3 ± 7.9% $n = 6$ (Fig. 2b), $p > 0.05$). Together with the almost complete abrogation of relaxation by URB597 (Fig. 1c) these data further confirm that COX acts downstream of AA in AEA-dependent airway relaxation. Of course we cannot rule out a contribution of direct COX-dependent metabolization of AEA completely. COX is known to generate the smooth muscle relaxants prostacyclin (PGI2) and prostaglandin E2 (PGE2) from AA and these compounds act via IP or EP receptors, respectively. Therefore, we then tested the contribution of these pathways to AEA-dependent airway relaxation. We found that IP, EP1, EP2, EP3, and EP4 receptors are all expressed in mouse trachea (Supplementary Fig. 2a). As activation of IP, EP2, and EP4 receptors has been already reported to mediate airway relaxation[12,13] we analyzed their role in AEA-dependent tone decrease in detail. The IP receptor blocker CAY10441 (3 μM) had almost no effect on AEA-dependent relaxation. In contrast, the EP2 receptor blocker AH6809 (10 μM) and especially the combination of AH6809 and the EP4 receptor blocker L161,982 (2 μM) strongly diminished airway relaxation by AEA (Fig. 2c). These results indicate that AEA induces airway relaxation at least in part by EP2 and EP4 receptor activation.

## AEA enhances PGE2 and cAMP production in airways

Since EP2/EP4-dependent airway relaxation by AEA is most likely induced by PGE2 we then quantified PGE2 generation in response to AEA application in trachea. For that purpose, tracheal rings were treated with AEA (10 μM) and the supernatant was collected after 0, 5, 15, and 30 min for ELISA measurements. We found increasing PGE2 concentrations in the supernatant over time resulting in a significant difference compared to solvent application after 30 min (Fig. 2d). Because PGE2 is known to evoke intracellular cAMP increases via EP2 and EP4 activation we have also assessed cAMP concentrations in tissue homogenates of AEA-treated tracheas. In these experiments we found elevated levels of cAMP 30 min after AEA application (10 μM) when compared to solvent control (Fig. 2e) as well. To identify the cell type/s generating PGE2 in trachea and to transfer our results to the human system we analyzed human tracheal epithelial cells (hTEPC) and human airway smooth muscle cells (hASMC). Immunostainings revealed FAAH expression in both cell types (Supplementary Fig. 2b, c). The specificity of the FAAH antibody was

shown in HUVECs using a lentiviral shRNA strategy (Supplementary Fig. 2d). PCR analysis demonstrated EP2 and EP4 receptor expression in hTEPC (Supplementary Fig. 2e) and hASMC (Supplementary Fig. 2f). Cells were treated with AEA (10 μM) for 30 and 60 min and then supernatants as well as cell homogenates were used for ELISA measurements. PGE2 levels were found to be elevated in the supernatant of hTEPC after 60 min (Fig. 2f) and in hASMC after 30 and 60 min (Fig. 2g) indicating that both cell types can generate PGE2 in response to AEA treatment. Moreover, cAMP levels were enhanced in hASMC 60 min after AEA administration (Fig. 2h) confirming that AEA induces the production of the relaxant molecule cAMP in this cell type. These results demonstrate that the AEA-induced signaling pathway we identified in mouse trachea is also relevant in cells from human airways. CAMP is also a mediator of the adrenergic signaling pathway that physiologically induces airway relaxation. To test if FAAH-dependent airway relaxation is regulated by critical physiological mediators of airway tone we incubated mouse tracheal rings for 1 h with the adrenergic bronchorelaxant isoproterenol (ISO, 10 μM) or the cholinergic bronchoconstrictor agonist methacholine (MCh, 10 μM). Quantitative PCR (qPCR) and activity measurements revealed that both compounds had neither an effect on FAAH expression (Supplementary Fig. 3a, b) nor FAAH activity (Supplementary Fig. 3c) in the airways. Then, we also examined a potential functional interaction of ISO and AEA signaling during long-term treatment. Therefore, tracheal rings were incubated for 18 h in medium supplemented with or without AEA (10 μM) or ISO (10 μM) and then analyzed in the wire-myograph. Our results demonstrate that prolonged treatment with ISO reduced subsequent relaxation by ISO (Fig. 2i) but not by AEA (Fig. 2j), while incubation with AEA neither affected ISO (Fig. 2k)- nor AEA (Fig. 2l)-dependent tone decrease. These findings illustrate that there is no functional interaction of beta adrenergic and AEA/FAAH signaling related to bronchorelaxation. Interestingly, in contrast to airway relaxation by beta agonists AEA/FAAH-dependent bronchorelaxation is not subject to desensitization during long-term treatment.

## AEA induces airway relaxation in precision-cut lung slices and reduces airway resistance in vivo

Then, we wanted to know if AEA also induces airway relaxation in smaller intrapulmonary airways. First, we analyzed FAAH expression along the airway tree and found increasing expression in higher generations of airways reaching similar levels as found in the brain, the positive control. Interestingly, FAAH was expressed much stronger in

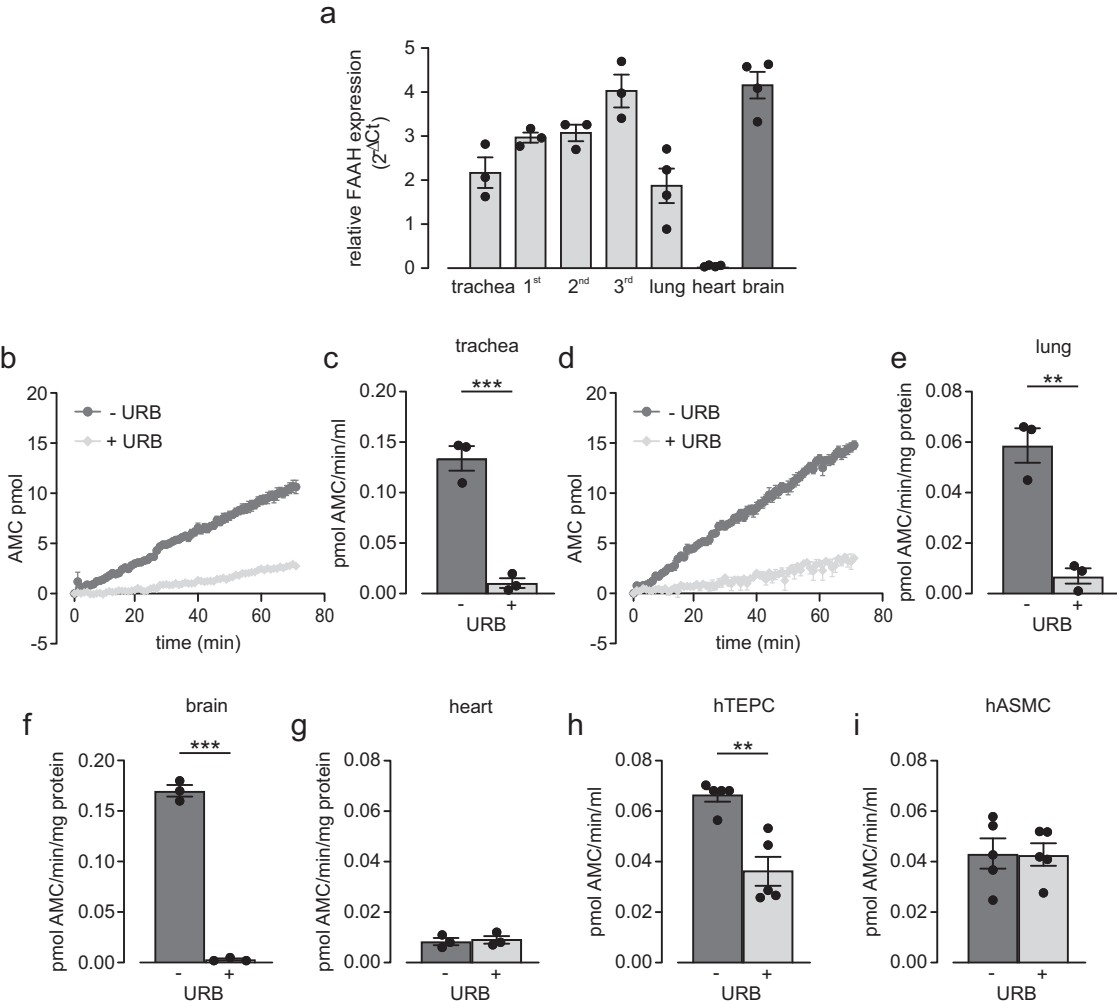

**Fig. 3 | FAAH activity can be detected in mouse lung. a** Analysis of FAAH
expression in different generations of C57BL/6J mouse airways (trachea, 1st, 2nd,
3rd airway generation $n = 3$), lung ($n = 4$), heart ($n = 4$), and brain ($n = 4$) demon-
strates high FAAH levels. Organs were derived from independent animals.
**b**, **c** FAAH activity can be found in C57BL/6J mouse trachea, original graph (**b**),
statistical analysis (**c**) ($n = 3$ independent animals), unpaired two-tailed Student's $t$
test ***$p = 7.0 \times 10^{-4}$. **d**, **e** FAAH activity can be also found in C57BL/6J mouse lung
tissue, original graph (**d**), statistical analysis (**e**) ($n = 3$ independent animals),

unpaired two-tailed Student's $t$ test **$p = 0.0023$). **f**, **g** Analysis demonstrates high
FAAH activity in mouse brain (positive control, **f**) but not in mouse heart (negative
control, **g**) ($n = 3$ independent animals), unpaired two-tailed Student's $t$ test
***$p = 3.3 \times 10^{-6}$ (**f**). **h**, **i** In human cells FAAH activity can be detected in hTEPC (**h**)
but not in hASMC (**i**) ($n = 5$ independent samples), unpaired two-tailed Student's $t$
test **$p = 0.0013$ (**h**); URB597 (URB, 10 μM). All data are presented as mean
values ± SEM, 7-amino-4-methylcoumarin (AMC). Source data are provided as a
Source Data file.

airways and lung when compared to the heart suggesting a functional
role in the lung (Fig. 3a). FAAH expression does not necessarily cor-
respond to FAAH activity. Therefore, we also examined FAAH activity
in lung tissue and cells and used again brain and heart as positive and
negative controls, respectively. Our data reveal that there is FAAH
activity in mouse trachea (Fig. 3b, c), lung (Fig. 3d, e) and in particular
in the brain (Fig. 3f) that can be abolished by the FAAH inhibitor
URB597 (URB, 10 μM). Interestingly, we could not detect FAAH activity
in the heart (Fig. 3g), which corresponds to low FAAH expression in this
organ. We also found FAAH activity in hTEPC (Fig. 3h) but not in
hASMC (Fig. 3i), which in the latter cell type is most likely due to FAAH
activity below the detection limit of the assay. Then, we tested the
functional effect of AEA on intrapulmonary airways. Therefore, we
applied precision-cut lung slices in which changes of airway lumen area
can be monitored by phase contrast microscopy. As expected ser-
otonin (5-HT, 0.1 μM) induced bronchoconstriction, which is reflected
by reduced lumen areas of intrapulmonary airways. After the addi-
tional application of AEA (10 μM, Fig. 4a, c) we found almost complete
relaxation of intrapulmonary airways, the solvent EtOH had no effect
(Fig. 4b, c). This response was not detected in lung slices derived from

FAAH$^{-/-}$ animals again confirming the involvement of FAAH-dependent
metabolites (Fig. 4c).

After we found that AEA induced relaxation in small intra-
pulmonary airways ex vivo we wondered if AEA may be also effective
in vivo. To test this we assessed the effect of AEA on airway mechanics
in anaesthetized and ventilated mice using the low-frequency forced
oscillation technique using the flexiVent system. To minimize potential
adverse effects we applied AEA (0.5 μg per mouse) to the lung via
inhalation as an aerosol. When comparing AEA with the solvent EtOH
these experiments demonstrate that starting from similar baseline
resistances AEA prevented the increase of airway resistance in
response to 5-HT (25 mg/ml) (Fig. 4d). For control, in another experi-
ment we applied the non-hydrolyzable analog Met-AEA (0.5 μg per
mouse) as an aerosol. As expected, this compound could not inhibit
the elevation of airway resistance in response to 5-HT indicating that
the bronchorelaxing effect of AEA is mediated via its FAAH-dependent
metabolites also in vivo (Supplementary Fig. 4a). Even though we
applied AEA to the lungs locally we wanted to rule out potential side
effects on the pulmonary circulation. To exclude that inhalation of AEA
affects pulmonary vascular pressure we monitored right ventricular

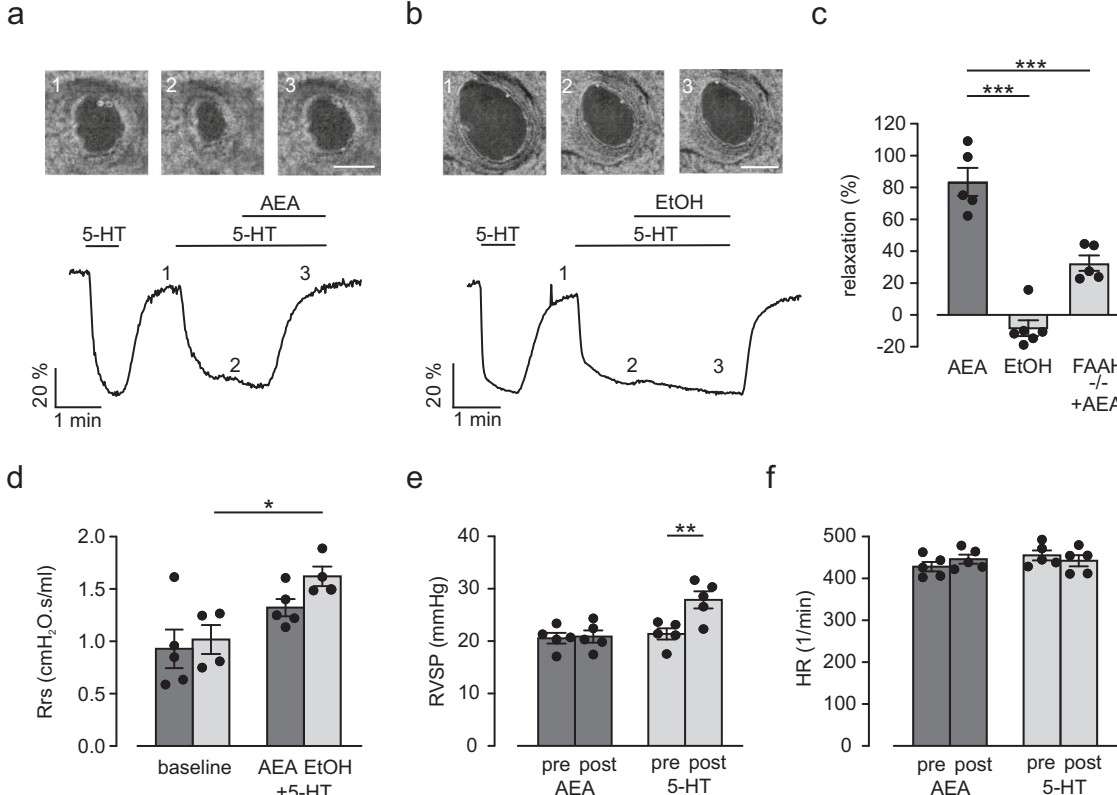

**Fig. 4 | AEA induces airway relaxation ex vivo and in vivo in healthy mice.**
**a**, **b** Top: phase contrast microscopy pictures of a small intrapulmonary C57BL/6J mouse airway. 1–3) Pictures represent time points during perfusion in the graph (bottom), scale bar = 50 μm. Bottom: Original trace of changes in airway lumen area. AEA (10 μM) completely reverses the reduction of lumen area by serotonin (5-HT, 0.1 μM) (**a**) while the solvent EtOH has no effect (**b**). **c** Statistical analysis reveals strong airway relaxation by AEA in intrapulmonary airways of C57BL/6J mice (AEA $n = 5$, EtOH $n = 6$ independent samples) that is attenuated in FAAH$^{-/-}$ mice ($n = 5$, independent samples), one way ANOVA, Tukey´s post hoc test (AEA vs EtOH ***$p = 3.2 \times 10^{-7}$; AEA vs AEA + FAAH$^{-/-}$ ***$p = 2.6 \times 10^{-4}$). **d** Analysis of airway resistance at baseline and after the subsequent inhalation of 25 mg/ml 5-HT together with AEA (0.5 mg per mouse, $n = 5$ independent animals) or the solvent EtOH ($n = 4$ independent animals) in healthy C57BL/6J mice. AEA limits the increase of airway resistance by 5-HT. Repeated measures two way ANOVA, Bonferroni's post hoc test *$p = 0.014$. **e** Analysis of right ventricular systolic pressure (RVSP) before (pre) and after (post) inhalation of AEA (0.5 mg per mouse) or 5-HT (50 mg/ml) as positive control in C57BL/6J mice ($n = 5$ independent animals); AEA inhalation has no effect on RVSP. Repeated measures two way ANOVA, Bonferroni's post hoc test **$p = 0.0026$. **f** Analysis of heart rate (HR) of C57BL/6J mice ($n = 5$ independent animals) before (pre) and after (post) inhalation of AEA (0.5 mg per mouse) or 5-HT (50 mg/ml); AEA and 5-HT have no effect on HR. All data are presented as mean values ± SEM. Source data are provided as a Source Data file.

pressure and heart rate using a small Millar catheter in the right ventricle of anaesthetized mice. AEA at the concentration found to reduce airway resistance (0.5 μg per mouse, Fig. 4d) was applied via a nebulizer. The analysis revealed that neither AEA nor the solvent EtOH affected right ventricular systolic pressure (RVSP) (Fig. 4e and Supplementary Fig. 4b) or heart rate (Fig. 4f and Supplementary Fig. 4c) thereby excluding adverse effects on the pulmonary circulation. As expected, positive controls demonstrated an increased RVSP by inhalation of the constrictor 5-HT (50 mg/ml) (Fig. 4e and Supplementary Fig. 4b), which confirmed the efficacy of aerosol application to the lung. Thus, inhalation of AEA prevents airway constriction in vivo without cardiovascular side effects.

**AEA levels and synthesis enzymes are reduced in the OVA-induced asthma model**
As AEA is known to be endogenously produced within the body, we wondered if the airway relaxing effect of AEA may also play a pathophysiological role in airway regulation e.g. in obstructive airway disease. To investigate this we have chosen the well-characterized OVA-induced asthma model. Mice were sensitized and then the levels of endocannabinoids and fatty acid amides were quantified in lung homogenates by LC-MRM. These experiments revealed reduced concentrations of AEA (Fig. 5a) and also of 2-arachidonylglycerol (2-AG) (Fig. 5b), AA (Fig. 5c), oleoylethanolamine (OEA) (Fig. 5d), and

palmitoylethanolamine (PEA) (Fig. 5e) in OVA-sensitized mice when compared to controls. We hypothesized that this may be due to reduced synthesis of the compounds. The main enzymes synthesizing AEA are N-acyl-phosphatidyl-ethanolamine phospholipase D (NAPE-PLD), a/b-hydrolase domain-containing protein 4 (ABDH4), glycerophosphodiesterase 1 (GDE1) and protein tyrosine phosphatase non-receptor type 22 (PTPN22). These all we found to be expressed in airway tissues and cells such as trachea (Supplementary Fig. 5a), hTPEC (Supplementary Fig. 5b) and hASMC (Supplementary Fig. 5c) even though to a different extent. We again determined activity of NAPE-PLD, the best-characterized biosynthetic enzyme of AEA, in the lung (Supplementary Fig. 5d) and the control organs brain (Supplementary Fig. 5e) and heart (Supplementary Fig. 5f). In all tissues examined we found NAPE-PLD activity that could be strongly reduced by the inhibitor combination of LEI-410 (LEI, 33 μM) and bithionol (Bith, 15 μM)[14,15] confirming the endogenous generation of AEA in particular in the lung. We also compared expression of these enzymes in the lungs of healthy and OVA-sensitized mice and found reduction of NAPE-PLD (Fig. 5f), ABDH4 (Fig. 5g), and GDE1 (Fig. 5h) but not PTPN22 (Fig. 5i) in asthmatic lung tissue. These results suggest that reduced expression levels of enzymes responsible for AEA synthesis in OVA asthma are related to diminished AEA tissue levels, which contributes to the hypercontractile phenotype of sensitized airways. Our endocannabinoid measurements in the lung revealed that also 2-AG levels were decreased in

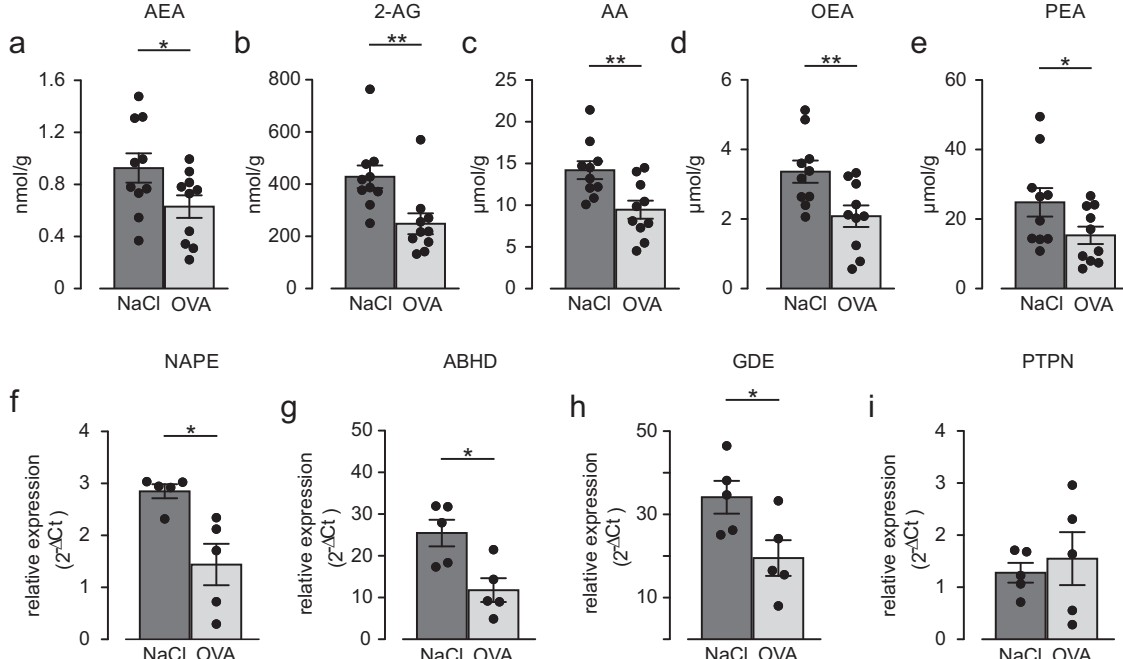

**Fig. 5 | Endocannabinoids as well as expression levels of enzymes promoting AEA synthesis are reduced in lungs of OVA-sensitized mice. a–e** Tissue levels of AEA (**a**), 2-Arachidonylglycerol (2-AG, **b**), AA (**c**), oleoylethanolamine (OEA, **d**), and palmitoylethanolamine (PEA, **e**) are reduced in OVA lungs of Balb/c mice ($n = 10$ independent animals). Unpaired two-tailed Student's $t$ test (AEA *$p = 0.026$; 2-AG **$p = 0.0034$; AA **$p = 0.003$; OEA **$p = 0.005$; PEA *$p = 0.032$). **f–i** Expression of N-acyl-phosphatidyl-ethanolamine phospholipase D (NAPE-PLD,

**f**), a/b-hydrolase domain-containing protein (ABDH4, **g**), glycerophosphodiesterase 1 (GDE1, **h**) but not protein tyrosine phosphatase nonreceptor type 22 (PTPN22, **i**) is diminished in OVA-sensitized Balb/c mice ($n = 5$ independent animals), unpaired two-tailed Student's $t$ test (NAPE *$p = 0.010$; ABDH4 *$p = 0.013$; GDE *$p = 0.037$). All data are presented as mean values ± SEM. Source data are provided as a Source Data file.

OVA-sensitized mice and 2-AG is also known to release AA upon hydrolysis. Therefore, we wondered if 2-AG can be a source of bronchorelaxing AA generation. To test this, we performed isometric force measurements in tracheal rings of mouse and found that 2-AG, in fact, induced a strong airway relaxation. However, this could neither be blocked by the monoacylglycerol lipase (MAGL) inhibitor MJN110 (MJN, 10 μM) nor the FAAH inhibitor URB597 (URB, 10 μM) (Supplementary Fig. 6a). To exclude that this is due to a lack of enzyme expression/activity of the 2-AG synthesis or degradation machinery in the lung we analyzed this and found DAGLalfa/beta expression (Supplementary Fig. 6b, c) as well as MAGL expression (Supplementary Fig. 6d) and activity (Supplementary Fig. 6e) in this organ, MAGL activity we also found in our control organs brain (Supplementary Fig. 6f) and heart (Supplementary Fig. 6g). These data indicate that airway relaxation by 2-AG is not mediated via AA; thus, bronchorelaxation via AA appears to be specifically induced by AEA.

## AEA induces airway relaxation and limits airway hyperresponsiveness in the OVA-induced asthma model

Due to diminished AEA tissue levels in asthmatic lungs we wondered if exogenous stimulation of the AEA/FAAH axis may also reduce airway tone in OVA asthma. First, we tested this in precision-cut lung slices ex vivo. Since our functional lung slice experiments shown in Fig. 4 had been exerted in C57BL/6 mice and OVA-dependent sensitization is mostly performed in Balb/c mice, we initially analyzed the effect of AEA in healthy Balb/c mice. As expected, also in this strain we found a prominent airway relaxation by AEA (Supplementary Fig. 6h, i). Then, we generated precision-cut lung slices of OVA-sensitized Balb/c mice. H&E stainings of cryosections confirmed the successful induction of acute and chronic asthma in these lungs by inflammatory cell invasion around the airways (Supplementary Fig. 6j–l). Next, we tested the functional effect of AEA on airway lumen area in precision-cut lung slices after induction of acute asthma. We found that AEA but not the

solvent EtOH induced airway relaxation to a comparable extent as in healthy Balb/c mice (Fig. 6a, b). Also in the chronic OVA model we found a strong relaxation of intrapulmonary airways by AEA (Fig. 6c, e) but not the solvent EtOH (Fig. 6e); moreover, the relaxation was similar to that in non-sensitized control animals receiving NaCl instead of OVA (Fig. 6d, e). This indicates that the airway relaxing potential of AEA is preserved in asthmatic animals. To assess the effect of AEA on airway hyperresponsiveness in vivo, we again determined airway mechanics using low force frequency measurements in OVA mice. First baseline resistance was recorded and then AEA or EtOH were applied together with 5-HT as an aerosol via a nebulizer. We found that baseline resistances were similar in both groups of animals. When 5-HT was applied the concomitant administration of AEA strongly reduced the enhancement of airway resistance by 5-HT in OVA-sensitized mice when compared to control (Fig. 6f). Thus, AEA is a prominent airway relaxant in asthmatic mice ex and in vivo.

## Discussion

Traditional Asian and African medicine applied cannabis in neurological disorders, infections and also lung disease such as bronchitis and asthma[16]. Also more recent reports suggest a potential beneficial effect of plant-derived THC in human asthmatics[17] even though also deleterious effects of marijuana smoking have been found. However, it is unclear which of the ingredients of the plant is responsible for these reactions. Because cannabinoids can be also endogenously produced in the body and have an important role in organ homeostasis, we investigated the impact of the endocannabinoid AEA on tone regulation in healthy airways and the well-characterized OVA-induced asthma model. Our ex and in vivo results demonstrate that AEA induces a powerful airway relaxation in health and disease that is mediated by FAAH-dependent metabolites. While FAAH was previously only considered as an enzyme that inactivates AEA, there is now accumulating evidence by us and others that also FAAH-dependent AEA metabolites

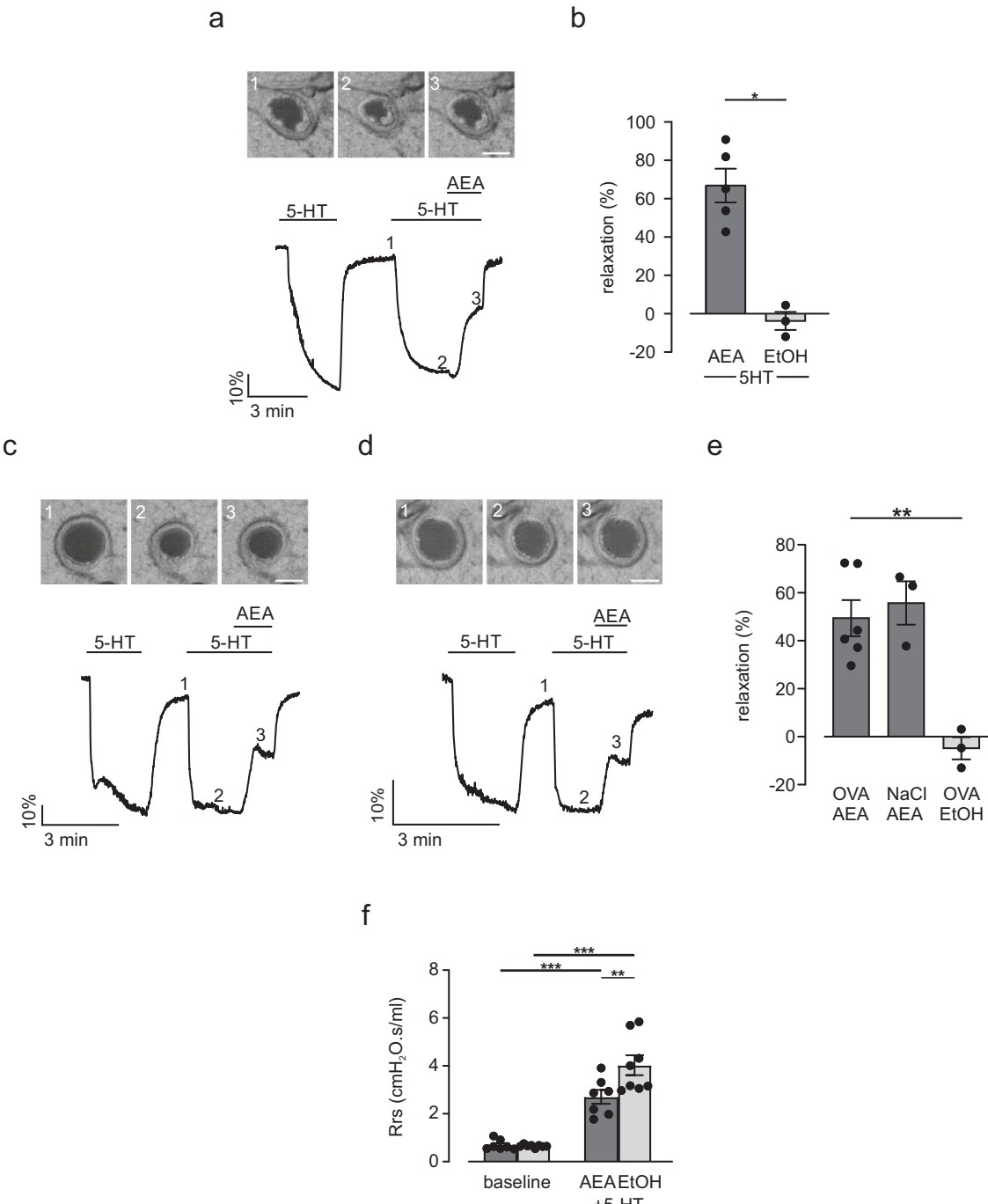

**Fig. 6 | AEA induces airway relaxation ex and in vivo in OVA-sensitized mice.**
**a** Top: phase contrast microscopy pictures of a small intrapulmonary Balb/c mouse airway. 1–3) Pictures represent time points during perfusion in the graph (bottom), scale bar = 50 μm. Bottom: Original trace of changes in airway lumen area. AEA (10 μM) reverses the reduction of lumen area by 5-HT (0.1 μM) in the acute OVA model. **b** Statistical analysis reveals strong airway relaxation by AEA (n = 5 independent animals) but not EtOH (n = 3 independent animals) in intrapulmonary airways of OVA-sensitized mice with acute asthma, unpaired two-tailed Student's t test *p = 0.012). **c, d** Top: phase contrast microscopy pictures of a small intrapulmonary Balb/c mouse airway. (1–3) Pictures represent time points during perfusion in the graph (bottom), scale bar = 50 μm. Bottom: original trace of changes in airway lumen area. AEA (10 μM) reverses the reduction of lumen area by 5-HT (0.1 μM) (**c**) in the chronic OVA model, a similar effect is found in controls without

asthma (NaCl) (**d**). **e** Statistical analysis reveals strong airway relaxation by AEA (n = 6 independent animals) but not EtOH (n = 3 independent animals) in intrapulmonary airways of OVA-sensitized Balb/c mice with chronic asthma as well as in control mice (NaCl, n = 3 independent animals), one way ANOVA, Tukey's post hoc test (OVA AEA vs OVA EtOH **p = 0.0025). **f** Analysis of airway resistance at baseline and after the subsequent inhalation of 25 mg/ml 5-HT together with AEA (0.5 mg per mouse, n = 7 independent animals) or the solvent EtOH (n = 8 independent animals) in Balb/c mice with acute OVA asthma. AEA limits the increase of airway resistance by 5-HT. Repeated measures two way ANOVA, Bonferroni's post hoc test (baseline vs AEA ***p = 3.6 × 10⁻⁴; baseline vs EtOH ***p = 8.7 × 10⁻⁷; AEA vs EtOH **p = 0.0032). All data are presented as mean values ± SEM. Source data are provided as a Source Data file.

initiate important signaling pathways[8,18]. We found very strong FAAH expression in murine lung, in particular in the airways. FAAH expression levels in the airways are much higher than in other peripheral organs (e.g., heart). Our RNA and protein expression analyses reveal that FAAH is found in mouse and also human airway smooth muscle and epithelial cells. The relevance of FAAH signaling in the lung was also corroborated by FAAH activity measurements in lung tissues and cells. These findings suggest a potential key role for FAAH in airway function. FAAH is known to metabolize AEA to arachidonic acid (AA) which is then further degraded via cyclooxygenase (COX), lipoxygenase (LOX), and cytochrome P450 enzymes to various eicosanoids[19]. Also AA proved to be a very strong airway relaxant in our experiments while the other FAAH-dependent metabolite ethanolamine induced a small bronchoconstriction. Importantly, in our myograph experiments there was a net bronchorelaxing effect of AEA degradation to AA and ethanolamine indicating that the airway relaxing effect of AA strongly outweighs ethanolamine-dependent constriction. Interestingly, with NAAA there is also an alternative enzyme known that can degrade AEA[20]. It has been shown to be strongly expressed and to be active in the lung[21,22]. However, it is mainly localized in alveolar macrophages[23] and therefore unlikely to regulate the tone of isolated healthy airways. In fact, our data show that NAAA inhibition does not affect AEA-dependent bronchorelaxation. A potential role of NAAA in asthmatic lung inflammation has to be investigated in future studies. Because we found that AEA evokes bronchorelaxation via its FAAH-dependent metabolite AA and because 2-AG can also release AA upon hydrolysis, we wondered if 2-AG can relax airways via its MAGL or FAAH-dependent degradation. However, our data suggest that 2-AG-dependent airway relaxation is not mediated via AA and future studies will have to characterize the underlying signaling pathway in detail. As direct mediator of AEA-induced airway relaxation we could identify PGE2 acting via EP2 and EP4 receptors. Accordingly, PGE2 has been reported to be a strong airway relaxant in human, mouse, pig, and monkey via these receptors before[12,24,25].

The AEA/FAAH pathway may be also of pathophysiological relevance in the lung as we found reduced levels of AEA and other fatty acid amides in the lungs of OVA-sensitized mice. The lack of these compounds and/or their metabolites could contribute to airway hyperreactivity in this disease. As a short term effect of allergen instillation in humans enhanced AEA levels were found in the BAL after 24 h[26]. In the light of our results this could be considered as an acute compensatory reaction in the lung to limit tone increase. In any case this finding confirms that AEA is involved in allergen-dependent signaling in human lungs. As a potential reason for the long-term reduction of AEA levels in OVA-sensitized mice in our experiments we identified reduced expression of enzymes responsible for AEA production such as NAPE-PLD, ABDH4, and GDE1[27]. This is in accordance with the reported diminished transcription of NAPE-PLD in response to inflammatory stimuli in macrophages[28]. While also other endocannabinoids may affect airway tone, the exogenous application of AEA or its FAAH-dependent metabolites can be an option to restore diminished AEA levels or metabolites to acutely counteract airway constriction. As a potential side effect, we have excluded pulmonary vasoconstriction of AEA inhalation in the lung at the concentration required for airway relaxation. Interestingly, similar to our finding for AEA also hypoxia has been reported to induce relaxation in airways[29,30]. In a previous study we could show that hypoxia evokes AEA production in the lung[8]. Hence it can be hypothesized that AEA is one of the mediators of this effect. Why AEA differentially regulates tone in pulmonary arteries and airways is currently unknown. One option may be different activities of COX, LOX and CYP450 enzymes in these tissues, but this aspect needs to be examined more in detail in future studies. Given that the AEA/FAAH pathway is not under control of beta adrenergic or cholinergic signaling it appears to be an additional endogenous airway tone modulating pathway with (patho)physiological relevance. This could

be also of therapeutic value in airway obstruction. However, future studies will have to assess which mediator (AEA and/or metabolite/s) is best suited for in vivo application. Moreover, potential effects of AEA/FAAH signaling on other features of asthma are of interest. While acute single dose application of AEA in the current study was not expected to affect inflammation and remodeling this should be investigated in detail in a future approach with chronic AEA administration. In addition, a potential impact of AEA/FAAH signaling on other types of obstructive lung disease such as COPD may be of therapeutic relevance. Thus, we have identified the AEA/FAAH axis as an important determinant of airway tone and AEA as a strong bronchorelaxant in large and small airways in health and disease.

## Methods

### Isometric force measurements of murine trachea
Isometric force measurements were performed as described previously[31,32]. In detail, mouse tracheas were isolated and in some tracheas the epithelium was mechanically removed. Then they were cut into rings in cold low-calcium PSS, containing 118 mM NaCl, 5 mM KCl, 1.2 mM $MgCl_2$, 1.5 mM $NaH_2PO_4$, 0.16 mM $CaCl_2$, 10 mM glucose, and 24 mM Hepes (pH 7.4). Rings were mounted on a wire myograph (Multi Myograph 620 M, Danish Myo Technology, Aarhus, Denmark) and pre-stretched to 5 mN in PSS, containing 118 mM NaCl, 5 mM KCl, 1.2 mM $MgCl_2$, 1.5 mM $NaH_2PO_4$, 1.6 mM $CaCl_2$, 10 mM glucose, and 24 mM Hepes (pH 7.4). After equilibration for 20 min maximal constriction was induced by 5-HT (10 μM, 8367.1, Carl Roth, Germany) or methacholine (MCh, 10 μM, A2251, Sigma-Aldrich, Germany). For analysis LabChart 8 (ADInstruments, Oxford, UK) was used. Forces were determined 3 min after drug application. These drugs were used: Anandamide (AEA, 1339, Tocris, Germany), arachidonic acid (AA, 90010, Cayman Chemical, USA), AH6809 (AH, 14050, Cayman Chemical, USA), 2-Arachidonylglycerol (2-AG,1298, Tocris, Germany), ARN726 (ARN, 24259, Cayman Chemical, USA), CAY10441 (CAY, 10005186, Cayman Chemical, USA), Indometacin (Indo, 70270, Cayman Chemical, USA), KCl (P9333, Sigma-Aldrich, Germany), L-161,982 (L, 10011565, Cayman Chemical, USA), MJN110 (MJN, 17583, Cayman Chemical, USA), Nordihydroguaiaretic Acid (NDGA, 70300, Cayman Chemical, USA), 17-Octadecynoic acid (ODYA, 90270, Cayman Chemical, USA), URB597 (URB, 10046, Cayman Chemical, USA).

### Short and long-term treatment of tracheal rings
For qPCR and FAAH activity assays short term incubation of tracheal rings (qPCR: 4 h, 37 °C; activity assay: 1 h, 37 °C) was performed in PSS with or without isoprenaline (ISO, 10 μM), acetyl-β-methylcholine chloride (MCh, 10 μM, A2251, Sigma-Aldrich, Germany) or URB597 (10 μM, 10046, Cayman Chemical, USA). In order to test a potential functional desensitization in response to ISO (10 μM) or anandamide (AEA, 10 μM, 1339, Tocris, Germany) long-term incubation of tracheal rings (18 h, 37 °C) was performed in Dulbecco's modified eagle´s medium (DMEM) containing 10% fetal calf serum (FCS), 1% penicillin/streptomycin (P/S), 1% non essential amino acids and 0.1% β-mercaptoethanol. After that tracheal rings were transferred to the wire-myograph and isometric force measurements were executed.

### Precision cut lung slices
Precision cut lung slices were generated as described previously[33,34]. In detail, mice were sacrificed, the trachea was cannulated and the lungs were filled with a warm solution of 4% low-melting point agarose (6351.5, Carl Roth, Germany) by use of a Saf-T-Intima catheter (Becton Dickinson GmbH, Germany). The agarose was flushed into the alveoli by a small volume of air. Warm gelatin solution (6%, G2500, type A, porcine skin; Sigma-Aldrich, Germany) was applied into the pulmonary vasculature via the right ventricle in order to fill small arteries. After gelling of agarose and gelatin at 4 °C in a fridge, lungs were removed and single lobes were separated. 200 μm thick lung slices were cut

using a vibratome (VT1200S, Leica, Germany) and then they were incubated over night at 37 °C in serum-free medium. The next day, slices were studied by perfusion with pharmacological compounds on the stage of an inverted microscope. Pictures of small intrapulmonary airways were taken with a CCD camera. Changes in lumen area of the small airways were determined by a custom-written software (Lumen Calc 2.4, National Instruments, Texas, USA).

## Cell culture

Human tracheal epithelial cells (hTEPC, SC-3220), human airway smooth muscle cells (hASMC, SC-3400), human umbilical vein endothelial cells (HUVEC, SC-8000) were obtained from Provitro (Provitro AG, Berlin, Germany). They were cultivated in medium supplied by the manufacturer. All cell lines were passaged once a week until passage 9. To detach the cells, accutase™ solution (hTEPC, HUVEC) (SCR005, STEMCELL Technologies Germany GmbH, Germany) or Trypsin/EDTA (hASMC) (0.05%, 25300-054, Thermo Fisher Scientific, USA) was used.

## Reverse transcription PCR

RT-PCR was performed as described before[35,36]. In detail, for reverse transcription PCR (RT-PCR) experiments total RNA of hTEPC, hASMC and mouse tracheas was extracted with the Direct-zol RNA Miniprep Kit (Zymo Research, Germany). CDNA was generated using the SuperScript VILO Kit (Invitrogen, USA). Following primers were applied: murine FAAH: 5′-CTCTGGGTTTAGGACCTGAC-3′ (forward) and 5′-GAGTGGGACTGGTGTAGTTG-3′ (reverse); murine CB1: 5′-TCGCTGCCTCTACCTTCTCC-3′ (forward) and 5′-AGGCCAGGCTCAACGTGAC-3′ (reverse); murine CB2: 5′-GGACAAGGCTCCACAAGAC-3′ (forward) and 5′-GCTGCTGATGAACAGGTACG-3′ (reverse); murine IP: 5′-CAGTCTCATGGCCCTGTTG-3′ (forward) and 5′-CACCCAGCTCCCTTCCTTAG-3′ (reverse); murine EP1: 5′-CCTCGTCTGCCTCATCCATC-3′ (forward) and 5′-GAAACCACTGTGCCGGGAAC-3′ (reverse); murine EP2: 5′-GAAGAAGCCGCTGCGGATTG-3′ (forward) and 5′-ACTGGCACTGGACTGGGTAG-3′ (reverse); murine EP3: 5′-CTTCGCTGAACCAGATCTTG-3′ (forward) and 5′-CTTCACAGGAACCAGCTAAC-3′ (reverse); murine EP4: 5′-GATGAACGGCCTCAGGTCAG-3′ (forward) and 5′-CTCAGGCCTCAGATGTTCAGG-3′ (reverse); murine GAPDH: 5′-GTGTTCCTACCCCCAATGTG-3′ (forward) and 5′-CTTGCTCAGTGTCCTTGCTG-3′ (reverse); human EP2: 5′-TTGTTCCACGTGCTGGTGAC-3′ (forward) and 5′-AGGATGGCAAAGACCCAAGG-3′ (reverse); human EP4: 5′-TCCTGCCAGACCTCTCACTG-3′ (forward) and 5′-GTCATAGTGGGGTAGCATCCG-3′ (reverse); human GAPDH: 5′-CCATCACTGCCACCCAGAAG-3′ (forward) and 5′-CCACCACCCTGTTGCTGTAC −3′ (reverse).

## Quantitative PCR

HTEPC and hASMC as well as mouse tracheas, lungs, hearts and brains were homogenized in TRIzol (Invitrogen, USA) using the TissueLyser LT (Qiagen, Germany). RNA was extracted with the Direct-zol RNA Miniprep Kit (hTEPC, hASMC, trachea, Zymo Research, Germany) or TRIzol reagent (lung, heart, brain). For cDNA generation the SuperScript VILO Kit (Invitrogen, USA) was used. For expression analysis QuantiTect Primer Assays (Qiagen, Germany) for ABDH4 (QT00112917), DAGLα (QT00167706), DAGLβ (QT00173453), FAAH (QT00149520), GDE1 (QT00160552), NAPE-PLD (QT00141729), MAGL (QT01163428), and PTPN22 (QT00103943) were applied. As housekeeper 18s rRNA (QT01036875) was used. QPCR was performed with the CFX96 Touch Real-Time PCR Detection System (Bio-Rad Laboratories GmbH, Hercules, CA, USA).

## PGE2 and cAMP ELISA

Mouse tracheas were isolated and cut into three rings in cold low-calcium PSS. HTEPC and hASMC were cultivated on 6-well plates until they reached 90% confluency. Then, tracheal rings or cells were incubated in PSS containing 3-Isobutyl-1-methylxanthin (IBMX, 10 μM, 15879, Sigma-Aldrich, Germany) for 30 min at 37 °C. After that, AEA

(10 μM, 1339, Tocris, Germany), ethanol or forskolin (positive control for cAMP, 10 μM, F6886, Sigma-Aldrich, Germany) were applied. For PGE2 measurements the supernatant was collected after further 5, 15 or 30 min (tracheal rings) or 30 or 60 min (cells). For cAMP measurements the same samples were used: Tracheal rings were lysed in HCl (0.1 M)/Triton X-100 (0.5%) with the help of two 7 mm stainless steel beads in the TissueLyser LT. Cells were also lysed with HCl/Triton X-100. Then, samples were centrifuged with 15,871×$g$ for 10 min at 4 °C and the supernatant was further treated with ultrasound in an ice-cold bath for 15 min. Total protein concentration of all samples was determined with the Pierce BCA Protein Assay Kit (23225, Thermo Fisher Scientific, USA). PGE2 was quantified in the supernatants using the Prostaglandin E2 ELISA Kit – Monoclonal (514010, Cayman Chemical, USA) and cAMP concentrations of tissue or cell lysates were determined with the Direct cAMP ELISA Kit (ADI-900-006, Enzo Life Sciences, Germany) according to the manufacturer's instructions.

## Histology

After isometric force measurements, tracheas were kept in Z-Fix for 24 h at 4 °C and then rinsed with water for 1 h. Tracheas were stored at 4 °C in 70% ethanol until paraffin embedding. In all, 10 μm thin slices were generated by a microtome (Leica, Germany) and H&E staining was performed. Tracheal sections were investigated with an Axiostar plus microscope and pictures generated with the AxioVision Rel. 4.8 software (Carl Zeiss, Germany).

## Activity assays of FAAH and MAGL

In order to determine FAAH or MAGL activity the FAAH activity assay kit (fluorometric) (ab252895, abcam, United Kingdom) or the MAGL activity assay kit (fluorometric) (ab273326, abcam, United Kingdom) was used according to manufacturer´s instructions. Mouse tissues (lung, trachea, brain and heart) were isolated, cut into small pieces and transferred into PSS with or without the FAAH inhibitor URB597 (10 μM, 10046, Cayman Chemical, USA) or the MAGL inhibitor MJN110 (10 μM, 17583, Cayman Chemical, USA). After incubation for 1 h at 37 °C tissues were transferred to 120 μl assay buffer with or without the inhibitors and lyzed with the help of two 7 mm stainless steel beads in a TissueLyser LT (Qiagen, Germany). Afterwards samples were kept on ice for 10 mins, centrifuged at 10,000×$g$ for 5 min at 4 °C and supernatant was transferred to a fresh tube. Total protein concentration was determined with the Pierce BCA Protein Assay Kit (Thermo Fisher Scientific, USA). For FAAH activity measurements of cells hASMC (50,000) and hTEPC (90,000) were seeded on a T-25 culture flask. After 6 days culture medium was changed to PSS with or without URB597 (10 μM, 10046, Cayman Chemical, USA) and the cells were incubated for 1 h. Then, cells were detached by a scraper, centrifuged and resuspendend in 120 μl assay buffer. After lysis in the TissueLyzer LT (see above) 50 μl of the supernatant was used for activity measurements. Fluorescence was recorded over time by a Tecan Infinite M1000 Pro reader (i-control software, Tecan Austria GmbH, Austria).

## NAPE-PLD surrogate substrate activity assay

The NAPE-PLD activity assay was performed in accordance to a previously described method with small changes[37]. Mouse tissues (lung, brain and heart) were isolated and cut into small pieces. These were incubated in PSS for 1 h (37 °C) with or without the NAPE-PLD inhibitor combination LEI-401 (LEI, 33 μM, 31108, Cayman Chemical, USA) and bithionol (Bith, 15 μM, 31622, Sigma-Aldrich, Germany) that was recently reported to provide prominent NAPE-PLD inhibition in HEK293 and HepG2 cells[14,15]. Then samples were transferred into assay buffer (50 mM Tris-HCl pH 7.5, 150 mM NaCl, and 0.02% v/v Triton X-100) containing the inhibitors and lyzed by 7 mm stainless steel beads in a TissueLyser LT. Supernatant was transferred to a fresh tube. A 1 mM stock solution of the substrate PED6

(Invitrogen, D23739) in DMSO was prepared and further diluted in DMSO (1/10) and assay buffer (1/10) to generate the 10 µM working solution. NAPE-PLD activity measurement was performed at a PED6 concentration of 1 µM in a final volume of 100 µl within a black Greiner 96-well plate (flat bottom). Fluorescence was measured using a Tecan Infinite M1000 Pro reader at 37 °C. Assay buffer with or without the inhibitors was used for background substraction. The enzymatic rate (relative fluorescence units (RFU)/min) was determined in the linear range of the curves and normalized to the protein content of the samples.

## Lentiviral transduction of HUVECs

Human Umbilical Vein Endothelial Cells (HUVEC) were seeded on glass cover slips in a 24 well plate at a density of 20,000 cells/well. Then, cells were transduced with shFAAH-containing lentivirus (MOI 2.5, TL313109V, Amsbio, Abingdon, UK) to downregulate FAAH. Native HUVEC and HUVEC transduced with control shRNA lentivirus served as negativ controls. After 72 h, cells were fixated and immunohistochemistriy was performed.

## Immunohistochemistry

HTEPC, hASMC or HUVEC were cultivated on glass cover slips until 75% confluency. Tracheal cryosections were generated by a cryotome. After permeabilization with Triton X-100 (0.2%, v/v) (Carl Roth, Germany) cells or sections were blocked with donkey serum (5%, w/v). For immunohistochemistry, primary antibodies against fatty acid amide hydrolase (FAAH) (sections: 1:50, 101600, Cayman) (1:100) and α-smooth muscle actin (asmac) (1:800, A5228, Sigma-Aldrich, Germany) were used. Secondary antibodies conjugated with Cy3 and Cy5 (1:400, 711-175-152, Jackson ImmunoResearch Laboratories, USA) were applied. For nuclear staining Hoechst 3342 (1:1000, B2261, Sigma-Aldrich, Germany) was used.

## Liquid chromatography-multiple reaction monitoring (LC-MRM) experiments of endocannabinoids

LC-MRM measurements were performed as described before[8]. In detail, frozen lung lobes were homogenized in cold extraction tubes with ceramic beads containing spiking solution (deuterated endocannabinoids in acetonitrile), 0.1 M formic acid (buffer) and ethylacetate/hexane (9:1). After homogenization with Precellys 24 samples were centrifuged and kept at −20 °C for 10 min to freeze the aqueous phase. Then, the upper organic phase was recovered, evaporated under a gentle stream of nitrogen at 37 °C and reconstituted in a solution of water/acetonitrile (1:1). The aqueous phase was used for determination of protein concentration. For LC-MRM, samples were injected and separated on a Phenomenex Luna 2.5-µm C18(2) HST column combined with a SecurityGuard precolumn (Phenomenex Inc., USA). Separated endocannabinoids were flow-through analyzed by MRM with the 5500 QTrap triple-quadrupole linear ion trap mass spectrometer armed with a Turbo V Ion Source (AB Sciex Germany GmbH, Germany). For quantification of the endocannabinoids in lung tissues, triplicate calibration curves were used. Protein concentrations of lung samples were determined using the Pierce BCA Assay Kit (Thermo Fisher Scientific, USA) and the endocannabinoid concentrations were normalized to total protein.

## Ovalbumin-induced asthma model

For acute asthma experiments, 8 weeks old female Balb/c (Janvier Labs, France) mice were sensitized on days 0 and 14 with 20 µg ovalbumin (OVA, A5503, Sigma-Aldrich, Germany) or NaCl (control) supplemented with 2 mg/ml Imject Alum (77161, Thermo Fisher Scientific, USA) by i.p. injection of 100 µl per mouse. On days 21, 22 and 23 the mice were challenged for 30 min by nebulization of 1% OVA (5 ml) or NaCl (control). On days 24 and 25 analysis was performed. For chronic asthma, OVA challenge was continued for 2 more weeks

(days 28–30, 35–37). Then, analysis was performed on days 38 and 39.

## Lung function measurements

Lung function experiments were performed with the flexiVent system (Scireq, Canada) using the Flexiware 8 software as described previously[38]. In detail, 30 min before anesthesia carprofen (5 mg/kg s.c.) was applied. Mice were anesthetized by an i.p. injection of fentanyl (50 µg/kg), medetomidine (0.5 mg/kg), and midazolam (5 mg/kg). As a muscle relaxant vecuronium (0.1 mg/kg) was used. After tracheotomy, the trachea was cannulated and mice were ventilated with a tidal volume of 10 ml/kg at 150 breaths/min and a positive end-expiratory pressure of 2.5 cmH$_2$O. An airway recruitment maneuver, consisting of two deep inflations, was performed before the measurements. Airway resistance was automatically calculated by the flexiWare 8 software. To challenge the airways, either a solution of AEA (0.5 µg/mouse, 10 µl, 1% EtOH in 0.9% NaCl) in 5-HT (25 mg/ml, 8367.1, Carl Roth, Germany), Met-AEA (0.5 µg/mouse, 10 µl, 1121, Tocris, Germany, 1% EtOH in 0.9% NaCl) in 5-HT (25 mg/ml) or the solvent ethanol in 5-HT (25 mg/ml) was applied as an aerosol via a nebulizer (AG-AL1100, Aerogen, Ireland).

## Right ventricular catheter measurements

Catheter measurements were performed as described previously[33]. Briefly, analgesia was induced by ketamine (50 mg/kg, i.p.) and xylazine (5 mg/kg, i.p.) and anesthesia was performed with 1.5% isoflurane (3% for induction, 1.5% during experiment). Then mice were intubated and ventilated by a rodent ventilator (Minivent, Hugo Sachs Elektronik, Germany). The thorax was opened and a Millar catheter (1 F) (Millar, USA) was inserted into the right ventricle. Pressure was recorded with the Millar Aria 1 system and either AEA (0.5 µg/mouse, 50 µl) or EtOH (1%, 50 µl) was applied as an aerosol via a nebulizer (AG-AL1000). For analysis, the LabChart 8 software was applied and pressure values at 2 min after application were determined. As positive control 5-HT (50 mg/ml) was nebulized at the end of each experiment. Only mice with a heart rate of more than 400 beats per min were analyzed.

## Animals

Animal experiments were performed in compliance with the guidelines of the German law and were approved by the LANUV (Landesamt für Natur, Umwelt und Verbraucherschutz NRW, Germany). For experiments 8- to 12-week-old female C57BL/6J (Charles River, Germany) and Balb/c (Janvier Labs, France) mice were used. Female FAAH$^{-/-}$ [39] and Cnr1/Cnr2$^{-/-}$ [40] mice (both C57BL/6 background) with an age of 8- to 12-weeks were kindly provided by the Institute of Molecular Psychiatry of the University Hospital Bonn. All mice were housed in standard cages on a 12 h light-dark cycle at 21 °C and 50% humidity and had ad libitum access to food and water. Mice were monitored on a daily basis. Euthanasia was performed by cervical dislocation, then, organs were removed and further processed.

## Statistical analysis

Statistical analysis was performed using GraphPad Prism 5.0 (GraphPad Software, San Diego, USA) or Excel 365 (Microsoft), and data are indicated as mean ± SEM. Figures were generated with Corel Draw 18 (Corel Corporation, Ottawa, CA). Wire myograph measurements were fitted to a sigmoidal dose-response curve with variable slope. Statistical differences of more than two experimental groups were determined by one or two way (repeated measures) ANOVA together with Tukey's or Bonferroni's post hoc test. For comparison of two groups, unpaired, two-tailed Student's $t$ test was used. Differences were considered significant when $P < 0.05$, $*p < 0.05$, $**p < 0.01$, $***p < 0.001$.

## Reporting summary

Further information on research design is available in the Nature Research Reporting Summary linked to this article.

## Data availability

All data supporting the findings from this study are available within the manuscript and its supplementary information. Source data are provided with this paper.

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

## Acknowledgements

We thank Caroline Geisen and Patricia Freitag (University of Bonn, Germany) for help with primer design and Andreas Zimmer (University of Bonn, Germany) for providing FAAH$^{-/-}$ and Cnr1/2$^{-/-}$ mice. We are grateful to Sarah Rieck (University of Bonn, Germany) for support with lentiviral transduction. In addition, we thank Claudia Schwitter (University of Mainz, Germany) and Stefanie Darnauer (University of Bochum, Germany) for excellent technical assistance. The project was funded by the Deutsche Forschungsgemeinschaft DFG (German Research Foundation, No. WE4461/1-1) to D.W. We acknowledge support by the DFG Open Access Publication Funds of the Ruhr-Universität Bochum.

## Author contributions

A.Si. performed myograph experiments, RT-PCR, ELISA, OVA asthma, precision-cut lung slice, and flexiVent experiments, she analyzed data and contributed to the writing of the manuscript, T.v.E. performed and analyzed myograph and precision cut lung slice experiments, A.Se. performed and analyzed catheter experiments, M.M. performed flexiVent experiments and immunostainings, L.B. performed LC-MRM experiments and analyzed these data, D.W. designed the study, supervised the experiments, and wrote the manuscript.

## Funding

## Competing interests

The authors declare no competing interests.
