## [Peer Review File · Nature Communications]

The endocannabinoid anandamide is an airway relaxant in health and diseaseEditorial Note: Parts of this Peer Review File have been redacted as indicated to maintain the confidentiality of unpublished data.

REVIEWER COMMENTS

Reviewer #1 (Remarks to the Author):

This study demonstrates that the endocannabinoid, anandamide promotes airway relaxation. The anandamide-induced airway dilation is independent of CB1 and CB2 receptors. Instead, a mechanism involving anandamide degradation by fatty acid amide hydrolase (FAAH) is proposed since pharmacological inhibitors of FAAH or FAAH^{-/-} mice display no airway relaxation in response to anandamide. The action of FAAH on anandamide results in the generation of PGE2 which is responsible for airway relaxation in both normal and allergic airways. It should be noted that the generation of bronchoactive prostanoids from anandamide by the action of FAAH and COX enzymes has been proposed previously by the authors Wenzel et al, in 2013 (ref 8) and by Shang et al., 2016 *Pharmacological Research*, 105, 152-163.

The strength of the study arises from determining the biological action of anandamide in the lung and resolution of the mechanism operative. This study reveals that in the lung a role for FAAH since immunostaining reveals that the enzyme is present in airway epithelium and smooth muscle of the trachea and high levels are present in the smaller airways. The airway relaxation is promoted by arachidonic acid but blocked by the COX inhibitor indomethacin, and inhibitors to EP2 and EP4 receptors block the effect. Consistent with this hypothesis anandamide treatment of trachea results in the generation of PGE2.

Interestingly, using an OVA-based model of allergic lung inflammation quantification of endocannabinoids found reduced levels of AEA, 2-AG and AA in lung tissues. It is proposed the reduced levels of endocannabinoids may contribute to the airway hyperactivity. Nevertheless, exogenous AEA reduced airway tone and reduced airway resistance and the effects of 5-HT in OVA treated mice.

These findings will have broad relevance to the field of airway function and asthma. Little attempt to establish whether anandamide mediated any immunoregulatory effects. Nevertheless the study is convincing and the results are presented in a compelling manner. The statistical analysis of the data seems appropriate throughout.

A minor observation is a preponderance of sentences starting with "because" - lines 153, 268, 277, 298, and 391. The occasional since or hence might help the flow.

Reviewer #2 (Remarks to the Author):

The manuscript by Dr. Wenzel's group clearly shows a so-called "AEA/FAAH axis" as a crucial regulator of airway tone, apparently via formation of PGE2 from arachidonic acid (AA) released from AEA upon hydrolysis by FAAH. Overall, I was surprised that an endocannabinoid like AEA (N-arachidonylethanolamine) could be used by our cells as a simple reservoir of AA, that in essence is what the authors are showing in this piece. In fact, endocannabinoids have attracted interest over the last decades for having their own biological activities, clearly distinct from those of AA (one of their two moieties with ethanolamine) and regulated by a rather complex "endocannabinoid system". Nevertheless, if in airway relaxation AEA does release AA, which then acts (as expected) as a relaxant upon conversion into PGE2, it seems all the more important to demonstrate that FAAH is indeed regulated by critical signals for airway tone. Such an evidence would support that the "AEA/FAAH axis" can complement the activity of authentic AA, maybe under specific (emergency) circumstances. This demonstration would represent a major step towards a better understanding of airway regulation, yet apparently not towards the use of local AEA as a therapeutic concept for airway relaxation: for this purpose direct application of AA seems much better. In addition to the above criticism, there are several flaws that should be addressed in order to improve the study, as detailed below.

1. Endocannabinoids do not have “a chemical structure similar to THC” (a terpeno-phenol compound), as wrongly stated at the beginning of the Introduction. Please amend this sentence.
2. If AEA acts as a reservoir of AA, also enzymes other than FAAH able to cleave it (e.g., NAAA) should be investigated. In addition, FAAH activity must be assayed to corroborate mRNA and protein expression in the experimental paradigms, because too often changes of these three parameters are not similar nor consistent.
3. Following up with the previous point, the main AEA biosynthetic enzymes (shown in Figure 4) must also be measured in the experimental paradigms used, and the effect of specific inhibitors (e.g., commercially available for NAPE-PLD) must be interrogated.
4. It seems relevant to show that the activity of AEA is unique for this endocannabinoid, and not for others that can also release AA upon hydrolysis: 2-arachidonoylglycerol (2-AG) must be tested in this context, and the expression and activity of its major metabolic enzymes (at the very least DAGL α /beta and MAGL) must be measured. Indeed, if the mechanism proposed for AEA is true, there is no obvious reason why 2-AG should not give the same effects upon cleavage and release of AA.
5. The authors should keep in mind that also AEA (as well as 2-AG) is a substrate for COX-2, LOX and CYP450 enzymes. Thus, when incubating AEA with inhibitors of these oxygenases, it cannot be ruled out that formation of oxidative products of AEA (rather than of AA) is prevented. The authors should improve their experiments to also block AEA hydrolysis in the presence of COX, 5-LOX and CYP450 inhibitors.
6. The authors should test also ethanolamine in their experiments, to rule out any effect of the other moiety of AEA.
7. All in vivo experiments with AEA should be performed also in the presence of Met-AEA as a negative control.
8. The suggestion of a potential therapeutic value of local AEA application should be deleted, because in fact AA should be used to this purpose.
9. Controls of the specificity of FAAH and NAPE-PLD antibodies should be provided in the immunohistochemistry, because these are highly debated tools.
10. Figure S1-C should be improved, to better show dose-dependence.

Rebuttal

Reviewer #1 (Remarks to the Author):

This study demonstrates that the endocannabinoid, anandamide promotes airway relaxation. The anandamide-induced airway dilation is independent of CB1 and CB2 receptors. Instead, a mechanism involving anandamide degradation by fatty acid amide hydrolase (FAAH) is proposed since pharmacological inhibitors of FAAH or FAAH^{-/-} mice display no airway relaxation in response to anandamide. The action of FAAH on anandamide results in the generation of PGE₂ which is responsible for airway relaxation in both normal and allergic airways. It should be noted that the generation of bronchoactive prostanoids from anandamide by the action of FAAH and COX enzymes has been proposed previously by the authors Wenzel et al, in 2013 (ref 8) and by Shang et al., 2016 *Pharmacological Research*, 105, 152-163.

The strength of the study arises from determining the biological action of anandamide in the lung and resolution of the mechanism operative. This study reveals that in the lung a role for FAAH since immunostaining reveals that the enzyme is present in airway epithelium and smooth muscle of the trachea and high levels are present in the smaller airways. The airway relaxation is promoted by arachidonic acid but blocked by the COX inhibitor indomethacin, inhibitors to EP₂ and EP₄ receptors block the effect. Consistent with this hypothesis anandamide treatment of trachea results in the generation of PGE₂.

Interestingly, using an OVA-based model of allergic lung inflammation quantification of endocannabinoids found reduced levels of AEA, 2-AG and AA in lung tissues. It is proposed the reduced levels of endocannabinoids is may contribute to the airway hyperactivity. Nevertheless, exogenous AEA reduced airway tone and reduced airway resistance and the effects of 5-HT in OVA treated mice.

These findings will have broad relevance to the field of airway function and asthma. Little attempt to establish whether anandamide mediated any immunoregulatory effects. Nevertheless the study is convincing and the results are presented in a compelling manner. The statistical analysis of the data seems appropriate throughout.

A minor observation is a preponderance of sentences starting with "because" - lines 153, 268, 277, 298, and 391. The occasional since or hence might help the flow.

Response: We thank the reviewer for the very positive comments.

1. The current study focuses on the acute effects of anandamide (AEA) on airway tone. The majority of the experiments are ex vivo or in vitro approaches with immune cells not present. In the in vivo experiments we applied single AEA doses and functional effects on airway or vascular tone were measured directly after AEA administration. Because of the acute application of AEA we did not expect to detect effects on the immune system. In fact, we analyzed bronchoalveolar lavage of OVA animals after single dose AEA application in Flexivent measurements in order to verify that allergic inflammation had developed by OVA pre-treatment before the experiment. In these OVA animals we found the typical increase of eosinophils but there were no differences in inflammatory cell composition irrespective if AEA or the solvent was applied. From this we conclude that single dose application of AEA does not acutely affect the immune system.

Potential effects of chronic AEA application on the immune system are beyond the scope of the current manuscript and will have to be investigated in future studies.

This was emphasized more strongly in the discussion section of the revised manuscript (p. 21, ll. 1518).

2. „Because“ was replaced in lines 69, 133, 144, 153, 268, 298, 300, 385 of the revised manuscript.

Reviewer #2 (Remarks to the Author):

The manuscript by Dr. Wenzel's group clearly shows a so-called "AEA/FAAH axis" as a crucial regulator of airway tone, apparently via formation of PGE2 from arachidonic acid (AA) released from AEA upon hydrolysis by FAAH. Overall, I was surprised that an endocannabinoid like AEA (N-arachidonylethanolamine) could be used by our cells as a simple reservoir of AA, that in essence is what the authors are showing in this piece. In fact, endocannabinoids have attracted interest over the last decades for having their own biological activities, clearly distinct from those of AA (one of their two moieties with ethanolamine) and regulated by a rather complex "endocannabinoid system". Nevertheless, if in airway relaxation AEA does release AA, which then acts (as expected) as a relaxant upon conversion into PGE2, it seems all the more important to demonstrate that FAAH is indeed regulated by critical signals for airway tone. Such an evidence would support that the "AEA/FAAH axis" can complement the activity of authentic AA, maybe under specific (emergency) circumstances. This demonstration would represent a major step towards a better understanding of airway regulation, yet apparently not towards the use of local AEA as a therapeutic concept for airway relaxation: for this purpose direct application of AA seems much better.

In addition to the above criticism, there are several flaws that should be addressed in order to improve the study, as detailed below.

Response: We thank the reviewer for her/his positive comments on our work and for providing constructive criticisms to improve our manuscript.

1. Endocannabinoids do not have "a chemical structure similar to THC" (a terpeno-phenol compound), as wrongly stated at the beginning of the Introduction. Please amend this sentence.

Response: We changed the sentence to „Endocannabinoids are endogenously-produced lipophilic compounds that can exert similar effects as tetrahydrocannabinol (THC), the main constituent of the cannabis plant.“

2. If AEA acts as a reservoir of AA, also enzymes other than FAAH able to cleave it (e.g. NAAA) should be investigated. In addition, FAAH activity must be assayed to corroborate mRNA and protein expression in the experimental paradigms, because too often changes of these three parameters are not similar nor consistent.

Response: We thank the reviewer for raising these important points.

In order to test if N-acylethanolamine acid amide hydrolase (NAAA), an enzyme that can act as an alternative hydrolase to cleave AEA to AA, is also involved in airway relaxation by AEA, we performed

isometric force measurements with murine tracheal rings in a wire-myograph. Similar to our experiments with the FAAH inhibitor URB597 we pre-incubated tracheal rings with the NAAA inhibitor ARN726 (ARN,

10 μ M), submaximal airway constriction was induced by 5-HT and then AEA was applied. The analysis revealed that the NAAA inhibitor could not block the relaxing effect of AEA (10 μ M) (AEA 88.6 \pm 4.3% n=7; ARN+AEA 81.2 \pm 3.4% n=5, p>0.05). This indicates that at least in healthy airways NAAA does not contribute to AEA hydrolysis and AA generation.

These data were included into the results section (p. 5, ll. 11-15) and Fig. S1I of the revised manuscript.

As suggested by the reviewer we analyzed FAAH activity in different lung tissues and cells by a fluorescence assay. Because in our FAAH expression analysis by qPCR (Fig. 3A) we used mouse brain and heart as positive and negative controls, respectively, we also determined FAAH activity in these tissues as controls. In order to corroborate that fluorescence increase in the assay is in fact a measure of FAAH activity, the FAAH inhibitor URB597 was always applied in control measurements. Our data reveal that there is FAAH activity in mouse lung, trachea and in particular in the brain that can be abolished by the FAAH inhibitor URB597. Interestingly, we could not detect FAAH activity in the heart, which corresponds to low FAAH expression in this organ. We also analyzed FAAH activity in human tracheal epithelial cells (hTEPC) and human airway smooth muscle cells (hASMC). Here we found that FAAH activity was inhibited by URB597 in hTEPC but we could not detect a URB-sensitive signal in hASMC. This can be most likely explained by very low FAAH activity in hASMC (below the detection limit of the assay). This is also corroborated by much lower PGE2 levels in these cells when compared to hTEPC (see ELISA measurements, Fig. 2 F, G).

We included FAAH activity measurements in the results section (p. 10, l.27-p.11, l.7), the discussion section (p. 20, ll.4-6) and new Fig. 3 B-I of the completely revised manuscript.

3. Following up with the previous point, the main AEA biosynthetic enzymes (shown in Figure 4) must also be measured in the experimental paradigms used, and the effect of specific inhibitors (e.g., commercially available for NAPE-PLD) must be interrogated.

Response: Our data show that the main biosynthetic enzymes of AEA, namely NAPE-PLD, ABHD4, GDE1 and PTPN22 are all expressed in trachea (new Fig. S5 A), hTEPCs (new Fig. S5 B), hASMC (new Fig. S5 C) and in the lung (Fig. 5 F-I) even though to different extents. Because NAPE-PLD is the best-characterized biosynthetic enzyme of AEA we also analyzed NAPE-PLD activity in the lung and control tissues (brain, heart). Therefore, we used a fluorescence assay with the specific NAPE-PLD inhibitor combination LEI-401 (LEI, 33 μ M) and bithionol (Bith, 15 μ M) that was recently reported to provide prominent NAPE-PLD inhibition in HEK293 and HepG2 cells^{1,2}. Our data reveal NAPE-PLD activity in the lung, brain and heart that can be strongly reduced by LEI+Bith (new Fig. S5 D-F) confirming the endogenous generation of AEA in the lung.

These new data were included into the results (p. 14, l.22 – p.15, l.4) section and new Fig. S5

(POINT 4 REDACTED ON AUTHORS' WISHES)

5. The authors should keep in mind that also AEA (as well as 2-AG) is a substrate for COX-2, LOX and CYP450 enzymes. Thus, when incubating AEA with inhibitors of these oxygenases, it cannot be ruled out that formation of oxidative products of AEA (rather than of AA) is prevented. The authors should improve their experiments to also block AEA hydrolysis in the presence of COX, 5-LOX and CYP450 inhibitors.

Response: We thank the reviewer for raising this interesting point.

When we block AEA hydrolysis by FAAH using URB597 the bronchorelaxing effect of AEA is completely abrogated (remaining relaxation similar to solvent ethanol). This suggests that oxygenation does not contribute to the effect. In order to provide additional evidence that reduced AEA-dependent bronchorelaxation by the COX inhibitor Indo (LOX and CYP450 inhibitors were almost without effect on AEA-induced relaxation) is AA-dependent, we determined the effect of Indo (10 μ M) on AA-induced bronchorelaxation in the wire-myograph. The results demonstrate that Indo reduces AA-induced relaxation to a similar extent as AEA-induced relaxation (Indo+AEA $33.9\pm 6.2\%$ n=7; Indo+AA $27.3\pm 7.9\%$ n=6, $p>0.05$). The complete abrogation of airway relaxation by FAAH inhibition together with the similar reduction of AEA- and AA-induced airway relaxation by Indo indicates that AEA-dependent airway relaxation is mediated via AA.

We included these data in the results section (p. 6, l.26 - p.7, l.4) and new Fig. 2B of the completely revised manuscript.

6. The authors should test also ethanolamine in their experiments, to rule out any effect of the other moiety of AEA.

Response: As suggested by the reviewer we have also tested ethanolamine in isometric force measurements. When applied on basal tone of mouse tracheal rings we found that ethanolamine (10 μ M) induces a small airway constriction (0.6 ± 0.4 mN, n=7). It has already been reported before that ethanolamine at very high concentrations of 10 mM can induce bronchoconstriction, presumably via unspecific activation of histamine and acetylcholine receptors ³. Importantly, in our myograph experiments with AEA there was a net bronchorelaxing effect of AEA degradation to AA and ethanolamine. This result suggests that the airway relaxing effect of AA outweighs a potential bronchoconstriction by ethanolamine.

The myograph results with ethanolamine were included into the results section (p. 6, ll. 17-21) and the effect of ethanolamine on airway tone is discussed (p. 20, ll. 8-12) in the revised manuscript.

7. All in vivo experiments with AEA should be performed also in the presence of Met-AEA as a negative control.

Response: Our in vivo analyses with the flexiVent revealed that AEA prevents the increase of airway resistance induced by 5-HT. According to our ex vivo/in vitro data this effect should be mediated by FAAH-dependent metabolites of AEA. As suggested by the reviewer, we now tested the non-hydrolyzable AEA analog Met-AEA as a negative control in mouse in vivo. Our data demonstrate that Met-AEA cannot prevent 5-HT-induced elevation of airway resistance, as expected. This result confirms that AEA-dependent metabolites are responsible for the airway relaxing effect of AEA also in vivo.

These new data are included in the revised results section (p. 12, ll. 21-24) and Fig. S4A of the manuscript.

8. The suggestion of a potential therapeutic value of local AEA application should be deleted, because in fact AA should be used to this purpose.

Response: We have deleted the suggestions of a potential therapeutic value of local AEA application, instead we refer to a potential therapeutic value of metabolites in the AEA/FAAH pathway (see discussion p. 21, ll. 1-2 and ll. 13-15), which includes AA.

0. Controls of the specificity of FAAH and NAPE-PLD antibodies should be provided in the immunohistochemistry, because these are highly debated tools.

Response: In our experiments we used a FAAH antibody to demonstrate protein expression of FAAH in mouse tracheal rings as well as hTEPC and hSMC. NAPE-PLD antibody was not used and therefore removed from the Methods section. In order to test the specificity of the FAAH antibody we analyzed FAAH expression in human umbilical vein endothelial cells (HUVEC) because these cells are well-known to express functional FAAH ⁴. We transduced the cells with shFAAH-containing lentivirus (TL313109V, Amsbio, Abingdon, UK) at a multiplicity of infection (MOI) of 2.5 to downregulate FAAH. Shctr RNA lentivirus served as a negative control. 3 days after transduction mRNA FAAH expression in HUVEC was reduced to about 55% and in a previous paper we could show that this protocol resulted in diminished angiogenesis ⁵. To test the FAAH antibody, 3d after transduction we fixated the cells and performed immunostainings of native HUVEC, HUVEC transduced with shctr lentivirus or shFAAH lentivirus. Immunofluorescence revealed that native HUVEC and HUVEC treated with shctr lentivirus showed strong FAAH expression while the signal was reduced in HUVEC after FAAH downregulation with shFAAH lentivirus. This confirms the specificity of the FAAH antibody applied.

These data were included into the results section (p. 8, ll. 2-3) and new Fig. S2D of the revised manuscript.

1. Figure S1-C should be improved, to better show dose-dependence.

Response: As suggested by the reviewer we have performed more dose-response experiments to better show dose-dependence of AEA-induced relaxation of mouse trachea and we have exchanged the original trace in Fig. S1C.

2. Investigation of potential FAAH regulation by critical signals for airway tone (see reviewer text above)

Response: As suggested by the reviewer we analyzed if FAAH is regulated by other mediators of airway tone. Airway tone is usually determined by the autonomic nervous system. Adrenergic mediators induce airway relaxation and cholinergic mediators evoke airway constriction. Therefore, we first investigated if the adrenergic agonist isoproterenol (ISO, 10 μ M) or the cholinergic agonist methacholine (MCh, 10 μ M) affect FAAH expression in airway tissue. Murine tracheal rings were incubated for 1 h with ISO or MCh and then FAAH expression was determined by qPCR. Our results demonstrate that treatment with ISO or MCh does not affect FAAH mRNA expression. Because enzyme expression does not necessarily correspond to enzyme activity, in another experiment we also incubated mouse lung tissue with ISO or MCh (10 μ M for 1 h before and during the assay) and measured FAAH activity with a fluorescence assay. To prove that the fluorescence signal in fact reflects FAAH activity, in some experiments we also applied the FAAH inhibitor URB597 (10 μ M) as a negative control. Also in this approach neither ISO nor MCh altered FAAH activity in the lung. These findings suggest that the AEA/FAAH signaling cascade is an airway relaxing pathway independent from adrenergic or cholinergic signaling. This is also underscored by our myograph experiments in Fig. 2 I-L

where over night incubation of tracheal rings with ISO had no effect on AEA-dependent airway relaxation and vice versa.

The new data were included into the results (p. 8, ll. 11-19 and ll. 23-27) and the discussion (p. 21, ll. 11-13) section and new Fig. S3 of the completely revised manuscript.

References

1. Aggarwal G, *et al.* Symmetrically substituted dichlorophenes inhibit N-acyl-phosphatidylethanolamine phospholipase D. *The Journal of biological chemistry* **295**, 7289-7300 (2020).
2. Zarrow JE, *et al.* Selective measurement of NAPE-PLD activity via a PLA1/2-resistant fluorogenic N-acyl-phosphatidylethanolamine analog. *Journal of lipid research* **63**, 100156 (2022).
3. Kamijo Y, Hayashi I, Ide A, Yoshimura K, Soma K, Majima M. Effects of inhaled monoethanolamine on bronchoconstriction. *Journal of applied toxicology : JAT* **29**, 15-19 (2009).
4. Maccarrone M, Bari M, Lorenzon T, Bisogno T, Di Marzo V, Finazzi-Agro A. Anandamide uptake by human endothelial cells and its regulation by nitric oxide. *The Journal of biological chemistry* **275**, 13484-13492 (2000).
5. Rieck S, *et al.* Inhibition of Vascular Growth by Modulation of the Anandamide/Fatty Acid Amide Hydrolase Axis. *Arteriosclerosis, thrombosis, and vascular biology* **41**, 2974-2989 (2021).

REVIEWER COMMENTS

Reviewer #2 (Remarks to the Author):

I found the revised version of this manuscript very much improved, due to the soundness of the new data. Yet, the following points remain to be addressed in order to make the study supportive of the conclusions.

Major points

1. To ascertain whether or not COX-generated metabolites of AEA (rather than of AA) are responsible for the observed bronchorelaxation, also treatment with INDO + URB must be performed. Using either inhibitor alone does not fully prove or disprove this point.
2. The effect of 2-AG should be tested also in the presence of INDO, in order to interrogate the possible activity of its COX-generated metabolites.

These two points seem pivotal to support the core message of the study, i.e. that AA released from AEA rather than oxygenated products of AEA itself (as well as of 2-AG), are responsible for the observed bronchorelaxation.

Minor point

It would be very informative to show that NAAA is indeed active, and that ARN treatment inhibits such an enzymatic activity.

Rebuttal

Reviewer #2 (Remarks to the Author):

I found the revised version of this manuscript very much improved, due to the soundness of the new data. Yet, the following points remain to be addressed in order to make the study supportive of the conclusions.

Major points

1. To ascertain whether or not COX-generated metabolites of AEA (rather than of AA) are responsible for the observed bronchorelaxation, also treatment with INDO + URB must be performed. Using either inhibitor alone does not fully prove or disprove this point.

Response:

We thank the reviewer for the very positive comments on our work.

We agree with the reviewer that the experiment with Indo + AEA alone does not tell us if the COX-dependent AEA metabolites that induce bronchorelaxation are derived from the FAAH/AA/COX pathway or the metabolization of AEA via COX directly. What makes us confident that FAAH/AA is involved is the observation that the FAAH inhibitor URB prevents AEA-dependent relaxation almost completely (see Fig. 1 C). If direct COX-dependent metabolites of AEA would be the main source of bronchorelaxants URB could not inhibit bronchorelaxation but would either have no effect or would rather increase bronchorelaxation via accumulation of AEA and subsequent induction of AEA degradation by COX. Having said this, as requested by the reviewer and to further confirm the important role of FAAH in AEA-induced bronchorelaxation we performed additional isometric force measurements with Indo + URB. We found that Indo could not further inhibit bronchorelaxation compared to URB alone (revised Fig. 2 A) indicating that COX acts downstream of AA. This together with the almost complete abrogation of AEA-dependent bronchorelaxation by URB and a similar inhibitory effect of Indo when used to block AEA- or AA-dependent bronchorelaxation (Fig. 2 B) indicates that AEA-dependent airway relaxation is mainly mediated via FAAH/AA. Of course we cannot rule out completely that also direct COX-dependent metabolization of AEA contributes to bronchorelaxation by AEA to a minor extent.

We have included the new data into the results section of the revised manuscript (p. 7, ll. 12-26) and revised Fig. 2A.

(POINT 2 REDACTED ON AUTHORS' WISHES)

3. Minor point

It would be very informative to show that NAAA is indeed active, and that ARN treatment inhibits such an enzymatic activity.

Response: Previous studies revealed that NAAA expression and enzyme activity are very high in the lung^{1, 2}. Interestingly, expression and activity appeared to be mainly restricted to pulmonary alveolar macrophages³. These data suggest that the bronchorelaxing effect of AEA in isolated trachea is unlikely to be mediated via NAAA as there are only very few macrophages in healthy airways⁴. In accordance with this notion our isometric force measurements on mouse tracheal rings show no effect of the NAAA inhibitor ARN726 (10 μ M) on AEA-dependent bronchorelaxation (Fig. S11); moreover, in our experiments AEA-induced bronchorelaxation was abrogated almost completely by the FAAH inhibitor URB597. NAAA has been reported to preferentially hydrolyze PEA and OEA⁵ thereby contributing to inflammation⁶. Thus, NAAA inhibitors have been successfully applied as therapeutic drugs in experimental models of inflammation in arthritis, dermatitis and nerve injury^{7, 8, 9, 6}. In fact, the NAAA inhibitor ARN726 was also applied in experimental lung inflammation; here, it inhibited NAAA activity and thereby increased PEA and OEA levels, which alleviated inflammatory responses in the models of carrageenan and LPS-induced lung inflammation^{10, 11}. Therefore, NAAA inhibitors could be promising compounds to attenuate inflammation also in asthma. However, to characterize the role of NAAA and its inhibitors in asthmatic lung inflammation is beyond the scope of the manuscript and needs to be further investigated in future studies.

We have included information on high expression and activity of NAAA in the lung and a potential role of this enzyme in lung inflammation of asthma into the discussion section of the revised manuscript (p. 21, ll. 13-18).

References

1. Tsuboi K, Sun YX, Okamoto Y, Araki N, Tonai T, Ueda N. Molecular characterization of N-acylethanolamine-hydrolyzing acid amidase, a novel member of the choloylglycine hydrolase family with structural and functional similarity to acid ceramidase. *The Journal of biological chemistry* **280**, 11082-11092 (2005).
2. Ueda N, Yamanaka K, Yamamoto S. Purification and characterization of an acid amidase selective for N-palmitoylethanolamine, a putative endogenous anti-inflammatory substance. *The Journal of biological chemistry* **276**, 35552-35557 (2001).
3. Tsuboi K, *et al.* Predominant expression of lysosomal N-acylethanolamine-hydrolyzing acid amidase in macrophages revealed by immunochemical studies. *Biochimica et biophysica acta* **1771**, 623-632 (2007).
4. Engler AE, *et al.* Airway-Associated Macrophages in Homeostasis and Repair. *Cell reports* **33**, 108553 (2020).
5. Romeo E, *et al.* Activity-Based Probe for N-Acylethanolamine Acid Amidase. *ACS chemical biology* **10**, 2057-2064 (2015).
6. Piomelli D, *et al.* N-Acylethanolamine Acid Amidase (NAAA): Structure, Function, and Inhibition. *Journal of medicinal chemistry* **63**, 7475-7490 (2020).

7. Bonezzi FT, *et al.* An Important Role for N-Acylethanolamine Acid Amidase in the Complete Freund's Adjuvant Rat Model of Arthritis. *The Journal of pharmacology and experimental therapeutics* **356**, 656-663 (2016).
8. Sasso O, *et al.* Antinociceptive effects of the N-acylethanolamine acid amidase inhibitor ARN077 in rodent pain models. *Pain* **154**, 350-360 (2013).
9. Sasso O, Summa M, Armirotti A, Pontis S, De Mei C, Piomelli D. The N-Acylethanolamine Acid Amidase Inhibitor ARN077 Suppresses Inflammation and Pruritus in a Mouse Model of Allergic Dermatitis. *The Journal of investigative dermatology* **138**, 562-569 (2018).
10. Ribeiro A, *et al.* A Potent Systemically Active N-Acylethanolamine Acid Amidase Inhibitor that Suppresses Inflammation and Human Macrophage Activation. *ACS chemical biology* **10**, 1838-1846 (2015).
11. Zhou P, Xiang L, Zhao D, Ren J, Qiu Y, Li Y. Synthesis, biological evaluation, and structure activity relationship (SAR) study of pyrrolidine amide derivatives as N-acylethanolamine acid amidase (NAAA) inhibitors. *MedChemComm* **10**, 252-262 (2019).

REVIEWERS' COMMENTS

Reviewer #2 (Remarks to the Author):

I found this newly revised manuscript very much improved and sound, due to the additional data that now make the study fully supportive of the conclusions.

I suggest that the different action of anandamide compared to 2-arachidonoylglycerol on bronchorelaxation is clearly stated in the title and abstract, even though the molecular details are not yet clear. The different action of these two prominent endocannabinoids is indeed a major outcome of this investigation. Maybe the new title could read: The endocannabinoids anandamide and 2-arachidonoylglycerol are airway relaxants in health and disease via different pathways

Rebuttal

Reviewer #2 (Remarks to the Author):

I found this newly revised manuscript very much improved and sound, due to the additional data that now make the study fully supportive of the conclusions.

I suggest that the different action of anandamide compared to 2-arachidonoylglycerol on bronchorelaxation is clearly stated in the title and abstract, even though the molecular details are not yet clear. The different action of these two prominent endocannabinoids is indeed a major outcome of this investigation. Maybe the new title could read: The endocannabinoids anandamide and 2-arachidonoylglycerol are airway relaxants in health and disease via different pathways.

Response:

We thank the reviewer for the positive comments on our manuscript.

As requested by the reviewer we have stated in the abstract that there are differential signaling mechanisms of bronchorelaxation induced by AEA and 2-AG.

We would prefer to keep the title unchanged, because we can only provide very few data on 2-AG-dependent signaling compared to rather extensive information on the mechanism of AEA-induced bronchorelaxation. Therefore, we believe that the title in its current form represents best the content of the manuscript.